# Structured Linear CDEs: Maximally Expressive and Parallel-in-Time Sequence Models

**Benjamin Walker[1], Lingyi Yang[1], Nicola Muça Cirone[2], Cristopher Salvi[2], Terry Lyons[1,2]**

[1]Mathematical Institute, University of Oxford
[2]Department of Mathematics, Imperial College London

## Abstract

This work introduces Structured Linear Controlled Differential Equations (SLiCEs), a unifying framework for sequence models with structured, input-dependent state-transition matrices that retain the maximal expressivity of dense matrices whilst being cheaper to compute. The framework encompasses existing architectures, such as input-dependent block-diagonal linear recurrent neural networks and DeltaNet's diagonal-plus-low-rank structure, as well as two novel variants based on sparsity and the Walsh–Hadamard transform. We prove that, unlike the diagonal state-transition matrices of S4D and Mamba, SLiCEs employing block-diagonal, sparse, or Walsh–Hadamard matrices match the maximal expressivity of dense matrices. Empirically, SLiCEs solve the $A_5$ state-tracking benchmark with a single layer, achieve best-in-class length generalisation on regular language tasks among parallel-in-time models, and match the performance of log neural controlled differential equations on six multivariate time-series classification datasets while cutting the average time per training step by a factor of twenty.

## 1 Introduction

Parallel-in-time architectures, such as Transformers and Structured State-Space Models (SSMs), have allowed language models to scale to billions of parameters [96, 39]. However, theory and practice agree that these models do not generalise to longer sequences on state-tracking problems, a task that classical Recurrent Neural Networks (RNNs) handle with ease [65, 66, 60]. Linear Neural Controlled Differential Equations (LNCDEs) are a continuous-time sequence model where the state-transition matrix, or vector field, depends linearly on the input path. This allows for the multiplicative interactions between the hidden state and the input path necessary for gating. Reframing SSMs as LNCDEs, it becomes clear that using a diagonal state-transition matrix severely restricts expressivity [22]. Replacing it with a dense matrix restores maximal expressivity and the ability to state-track, but increases the number of parameters and computational cost from $\mathcal{O}(d_h^2)$ to $\mathcal{O}(d_h^3)$, where $d_h$ is the hidden dimension [66, 22].

Structured alternatives seek the best of both worlds. Block-diagonal Linear RNNs (LRNNs) [33] and the Diagonal-Plus-Low-Rank (DPLR) structure of DeltaNet [85, 101, 88] reduce computational cost while preserving some expressivity, with the latter recently shown to be maximally expressive [72]. We generalise and extend these ideas with Structured Linear Controlled Differential Equations (SLiCEs), a unifying framework for structured, input-dependent state-transition matrices. SLiCEs replace the dense state-transition matrix of an LNCDE with an efficient structured variant, such as block-diagonal, sparse, Walsh–Hadamard or DPLR, which maintain the maximal expressivity of dense matrices whilst reducing both the parameter count and computational cost.

39th Conference on Neural Information Processing Systems (NeurIPS 2025).

**Contributions**

1. Structured Linear Controlled Differential Equations (SLiCEs) are introduced as a common framework for models with structured, input-dependent state-transition matrices. SLiCEs incorporate SSMs such as Mamba [39], LRNNs such as DeltaNet [85] and input-dependent block-diagonal LRNN [33], LNCDEs [22], and two novel structures based on sparse matrices and the Walsh–Hadamard transform.

2. Block-diagonal, sparse, and Walsh–Hadamard SLiCEs are proven to achieve maximal probabilistic expressivity in Theorems 4.1, 4.2, and 4.3, respectively. Previously, such expressivity had only been shown for dense and DPLR matrices [22, 72].

3. A comprehensive empirical evaluation showing that the structure of the state-transition matrix significantly impacts length generalisation on state-tracking problems. The block-diagonal structure emerges as a promising option, due to its strong empirical results and favourable parallelisation. Furthermore, on six real-world multivariate time-series classification datasets, a block-diagonal SLiCE is shown to match the predictive accuracy of Log-NCDEs, whilst reducing the per-step training time by a factor of twenty.

4. Open-source implementations of SLiCEs in both PyTorch and JAX, along with code to fully reproduce all experiments from this paper. These are available at `https://github.com/Benjamin-Walker/structured-linear-cdes` (PyTorch) and `https://github.com/Benjamin-Walker/log-neural-cdes` (JAX).

**Related work**  Increasing the expressivity of parallel-in-time sequence models while retaining their computational efficiency is of significant interest, as it would facilitate training large, performant models. One approach parallelises non-linear RNNs by rewriting them as fixed-point problems and applying parallel Newton or quasi-Newton methods to calculate their output [59, 36]. However, reported wall-clock gains remain limited; parallel autoregressive generation can be up to twice as slow as sequential baselines [36]. There have also been a large number of input-dependent LRNN architectures proposed, including input-dependent block-diagonal LRNN [33], DeltaNet [100], DeltaProduct [88], Gated DeltaNet [101], Mamba [39], Mamba-2 [28], RWKV-7 [76], HGRN-2 [79], mLSTM [6], Gated Linear Attention [99], Gated Random Feature Attention [77], Gated Slot Attention [102], TTT-Linear [92], and Titans [7]. These models use either diagonal or DPLR state-transition matrices. Table 2 in [100] presents a comparison of the architectures for a number of these models.

Utilising structured matrices to reduce the computational burden of neural networks extends beyond sequence models. The lottery-ticket hypothesis [35] argues that dense networks contain sparse sub-networks that, when trained in isolation, can match the accuracy of the full model. Such sub-networks have been uncovered by pruning before [94], during [91], and after training [43]. SLiCEs differ from pruning by imposing structured sparsity at initialisation and training the resulting sparse model directly. Other structured-matrix approaches include sparse Transformers [16], 2:4 sparsity in linear layers [67], and Monarch layers, which factorise weight matrices into two block-diagonal components [29].

## 2 Background

### 2.1 Linear controlled differential equations

Let $\omega : [0, T] \to \mathbb{R}^{d_\omega}$ be a path with bounded-variation, where $\omega_s$ denotes the value of the path at time $s \in [0, T]$. A linear Controlled Differential Equation (CDE) takes the form

$$\mathrm{d}h_s = \sum_{i=1}^{d_\omega} A^i h_s \mathrm{d}\omega_s^i, \tag{1}$$

where $A^i \in \mathbb{R}^{d_h \times d_h}$ is the linear vector field for each channel $i = 1, \ldots, d_\omega$, and $h : [0, T] \to \mathbb{R}^{d_h}$ is the solution path. Approximating $\omega_s$ with linear interpolation on the grid $0 = t_0 < \cdots < t_n = T$ yields

$$\tilde{h}_{t_{j+1}} = \exp\left(\sum_{i=1}^{d_\omega} (\omega_{t_{j+1}}^i - \omega_{t_j}^i) A^i\right) \tilde{h}_{t_j}. \tag{2}$$

The outputs $\tilde{h}_{t_j}$ are computable in $\mathcal{O}(\log(n))$ parallel steps by composing the flow on each interval using an associative parallel scan [9]. This approach has been used to parallelise linear RNNs [64] and SSMs [89].

On each interval $[t_j, t_{j+1}]$, (2) uses only the increments of $\omega$, providing a first-order approximation. The Log-ODE method extends this to higher orders by combining the iterated Lie brackets of the vector field with the log-signature of $\omega$ [12]. See Cass and Salvi [11, Section 3.2.2] for a summary description of the algorithm and Appendix C for a description of applying the algorithm to an LNCDE.

## 2.2 Linear neural controlled differential equations

Let $\{(t_i, x_i)\}_{i=0}^n$ denote a set of observations from a multivariate time-series and $X : [t_0, t_n] \to \mathbb{R}^{d_X}$ be a continuous interpolation, such that $X_{t_i} = (t_i, x_i)$. NCDEs are defined as

$$h_{t_0} = \xi_\phi(t_0, x_0), \quad h_t = h_{t_0} + \int_{t_0}^t g_\theta(h_s) \mathrm{d}X_s, \quad z_t = l_\psi(h_t), \tag{3}$$

where $\xi_\phi : \mathbb{R}^{d_X} \to \mathbb{R}^{d_h}$ and $g_\theta : \mathbb{R}^{d_h} \to \mathbb{R}^{d_h \times d_X}$ are neural networks, and $l_\psi : \mathbb{R}^{d_h} \to \mathbb{R}^{d_z}$ is a linear map [53]. NCDEs have a number of desirable properties, including maximal expressivity and robustness to irregular sampling rates. Building on the work of Neural Rough Differential Equations [70], Log-NCDEs [98] demonstrate that combining NCDEs with the Log-ODE method during training leads to state-of-the-art performance on a range of multivariate time-series modelling benchmarks with up to 50,000 observations.

LNCDEs take the form

$$h_t = h_{t_0} + \int_{t_0}^t \sum_{i=1}^{d_\omega} A_\theta^i h_s \mathrm{d}\omega_s^{X,i} = h_{t_0} + \int_{t_0}^t \left( \sum_{i=1}^{d_\omega} A_\theta^i \mathrm{d}\omega_s^{X,i} \right) h_s, \tag{4}$$

where $\omega^X : [t_0, t_n] \to \mathbb{R}^{d_\omega}$ is a path which depends on the input $X$ and the $A_\theta^i$ are trainable matrices. As will be discussed further in Section 3.2, LNCDEs are maximally expressive [53, 22]. Therefore, there exists a maximally expressive sequence model whose recurrence can be calculated parallel-in-time using the approach outlined in Section 2.1. However, the number of parameters and computational cost makes this approach infeasible in large models. Independently of LNCDEs, Merrill et al. [66] proposed IDS4, a modification of the S4 layer designed to allow state-tracking, which has the same form as (4). Appendix A provides a more detailed introduction to LNCDEs by comparing and contrasting them to SSMs and LRNNs. Additionally, Appendix A contains a toy example demonstrating how the structure of the matrices $A_\theta^i$ affects model expressivity, a discussion of how to extend LNCDEs to matrix-valued hidden states, and a pseudo-code implementation.

## 3 Expressivity

### 3.1 Introduction

Expressivity characterises the set of functions a model can approximate, and maximal expressivity (or universal approximation) guarantees that, with suitable parameters, any continuous function on a compact set can be approximated arbitrarily closely.

**Definition 3.1** (Maximal expressivity). *Let $\mathcal{X}$ be a topological space, and let $\mathcal{F} = \{f_\theta : \mathcal{X} \to \mathbb{R} \mid \theta \in \Theta\}$ be a class of real-valued functions on $\mathcal{X}$, parametrised by some set $\Theta$. We say that $\mathcal{F}$ is maximally expressive (or universal) if, for every compact set $\mathcal{K} \subset \mathcal{X}$ and every real-valued continuous function $f : \mathcal{K} \to \mathbb{R}$, the following property holds:*

$$\forall \epsilon > 0, \ \exists \theta \in \Theta \quad s.t. \quad \sup_{x \in \mathcal{K}} \left| f(x) - f_\theta(x) \right| < \epsilon. \tag{5}$$

A classical result is the Universal Approximation Theorem, which states that for $\mathcal{X} = \mathbb{R}^d$, single-hidden-layer feed-forward networks with a suitable activation function are maximally expressive [27, 47].

## 3.2 Maximally expressive models on paths

Let $\mathcal{X}$ denote the space of continuous paths of bounded variation on the interval $[0, T]$ that start at the same point and contain time as a channel (time-augmented). We endow this space with the $1-$variation topology. Let $\mathcal{F}$ be the class of LNCDEs defined in (4) with a linear readout layer $l_{\theta_2}$, such that $f_\theta : \mathcal{X} \to \mathbb{R}$ is defined by

$$\omega \mapsto l_{\theta_2}(h_{t_n}) = l_{\theta_2}\left(h_{t_0} + \int_{t_0}^{t_n} \sum_{i=1}^{d_\omega} A_{\theta_1}^i h_s \mathrm{d}\omega_s^i\right), \tag{6}$$

for $\omega \in \mathcal{X}$. In this setting, $\mathcal{F}$ is maximally expressive [51]. Furthermore, LNCDEs with diagonal matrices $A_{\theta_1}^i$, such as S4D [40] and Mamba [39], are not maximally expressive [22]. An alternative to maximal expressivity is the following probabilistic property.

**Definition 3.2** (Maximal Probabilistic Expressivity). *Let $\mathcal{X}$ be a topological space, $N \in \mathbb{N}$, and $\mathcal{F}^N = \{f_\theta^N : \mathcal{X} \to \mathbb{R} \mid \theta \in \Theta^N\}$ be a class of real-valued functions on $\mathcal{X}$ defined by*

$$f_\theta^N(\omega) = l_{\theta_2}(\tilde{f}_{\theta_1}^N(\omega)), \tag{7}$$

*for $\omega \in \mathcal{X}$, where $\tilde{f}_{\theta_1}^N : \mathcal{X} \to \mathbb{R}^N$, $l_{\theta_2} \in \mathbb{R}^N$ is a linear readout, $\theta_1 \in \Theta_1^N$, $\theta_2 \in \Theta_2^N$, and $\Theta^N = \Theta_1^N \cup \Theta_2^N$. Given a sequence of probability measures $\mathbb{P}_N$ on $\Theta_1^N$ with $\theta_1 \sim \mathbb{P}_N$, $\mathcal{F}$ has maximal probabilistic expressivity if, for every compact set $\mathcal{K} \subset \mathcal{X}$ and every real-valued continuous function $f : \mathcal{K} \to \mathbb{R}$, the following property holds:*

$$\forall \epsilon > 0, \quad \lim_{N \to \infty} \mathbb{P}_N \left\{ \exists l_{\theta_2} \ s.t. \ \sup_{\omega \in \mathcal{K}} \left| f(\omega) - f_\theta^N(\omega) \right| < \epsilon \right\} = 1. \tag{8}$$

In the context of machine learning, maximal probabilistic expressivity may be considered a more promising property than maximal expressivity, as for large enough $N$, it implies there exists a significant abundance of parameters $\theta_1$ that are capable of achieving uniformly bounded and arbitrarily low error rates with a linear readout layer. This suggests the parameters should be readily discoverable through standard optimisation methods.

In the case of LNCDEs, $N = d_h$,

$$\tilde{f}_{\theta_1}^{d_h}(\omega) = h_{t_0} + \int_{t_0}^{t_n} \sum_{i=1}^{d_\omega} A_{\theta_1}^i h_s \mathrm{d}\omega_s^i, \tag{9}$$

and $\mathbb{P}_{d_h}$ on $\Theta_1^{d_h}$ is a collection of probabilities on matrices $\mathbb{P}_{d_h}^i$ with $A_{\theta_1}^i \sim \mathbb{P}_{d_h}^i$. Achieving maximal probabilistic expressivity depends crucially on the choice of $\mathbb{P}_{d_h}^i$. Building on the work of Cuchiero et al. [26], Cirone et al. [22, Theorem B.13] showed that choosing the $A_\theta^i$ to be dense Gaussian matrices with independent entries achieves maximal probabilistic expressivity. Unfortunately, using dense matrices is infeasible in practice due to computational constraints, as discussed in Section 2.2.

## 4 Structured linear controlled differential equations

### 4.1 Introduction

Building on Cirone et al. [22], we introduce Structured Linear Controlled Differential Equations (SLiCEs), a unifying framework for various choices of structured $A_\theta^i$. This section presents three examples based on current models in the literature, diagonal, DPLR, and block-diagonal matrices, as well as two novel examples based on sparse matrices and the Walsh–Hadamard transform. Here, we focus on the recurrent core, with further implementation details given in Appendix A.3.

Our main theoretical results are Theorems 4.1, 4.2, and 4.3, which show that SLiCEs with block-diagonal, sparse, and Walsh–Hadamard matrices have maximal probabilistic expressivity. Formal proofs are given in Appendix B. These results complement [22, Theorem 4.3], which shows that SLiCEs with diagonal matrices are not maximally expressive and [72, Proposition F.2], which shows that SLiCEs with DPLR matrices have maximal probabilistic expressivity.

## 4.2 Diagonal-plus-low-rank SLiCEs

SSMs with diagonal state-transition matrices, such as S4D [40] and Mamba [39], are examples of SLiCEs with diagonal matrices, $A_\theta^i = D_\theta^i$. Hence, they are not maximally expressive and underperform on state-tracking benchmarks [22, 66]. This limited expressivity motivates the use of alternative structured state-transition matrices.

DeltaNet, DeltaProduct, and Gated DeltaNet use specific versions of DPLR state-transition matrices [85, 100, 88, 101]. The general form of a DPLR-SLiCE is

$$A_\theta^i = D_\theta^i + \sum_{j=1}^r u_\theta^{i,j} (v_\theta^{i,j})^\top, \tag{10}$$

where $r$ is the rank. This parameterisation reduces the number of trainable parameters and computational cost of calculating a hidden state update from $\mathcal{O}(d_\omega d_h^2)$ to $\mathcal{O}(d_\omega r d_h)$. Furthermore, [72, Proposition D.2] shows that if $r \to \infty$ as $d_h \to \infty$, then DPLR-SLiCEs have maximal probabilistic expressivity.

## 4.3 Block-Diagonal SLiCEs

Block-diagonal state-transition matrices were first explored in LRNNs to improve performance on regular-language tasks [33]. Block-diagonal SLiCEs (BD-SLiCEs) use the same structure, but make the dependence on the input path linear,

$$A_\theta^i = \text{BlockDiag}(B_{\theta,1}^i, B_{\theta,2}^i, \dots, B_{\theta,k}^i), \tag{11}$$

where each $B_{\theta,j}^i \in \mathbb{R}^{b_j \times b_j}$ is a trainable dense block, $k$ is the number of blocks, and $b_j$ are the block-sizes, with $d_h = \sum_{j=1}^k b_j$. This parameterisation reduces the number of trainable parameters and computational cost of calculating a hidden state-update from $\mathcal{O}(d_\omega d_h^2)$ to $\mathcal{O}(d_\omega \sum_{j=1}^k b_j^2)$, providing a substantial speed-up when each $b_j \ll d_h$. Furthermore, this does not restrict the expressivity.

**Theorem 4.1.** *If $\max_j b_j \to \infty$ as $d_h \to \infty$, then block-diagonal SLiCEs have maximal probabilistic expressivity.*

Hence, the non-linear dependence of the input-dependent block-diagonal LRNN is not necessary for theoretical expressivity. Because the hidden state factorises into $k$ independent parts, BD-SLiCEs can be viewed as a multi-head dense LNCDE (DE-LNCDE) of head sizes $b_j$. For a fixed $d_h$, choosing smaller blocks yields greater speed; choosing larger blocks yields greater expressivity. Under a fixed compute budget, there is a trade-off between expressivity and hidden dimension, and this is explored empirically in Appendix D.2.

## 4.4 Sparse SLiCEs

Let $0 < \epsilon < 1$. A sparse SLiCE (S-SLiCE) takes each $A_\theta^i$ to be a sparse matrix with $\mathcal{O}(d_h^{1+\epsilon})$ non-zero entries, selected at random according to a Bernoulli distribution. This reduces the parameter count and computational cost of calculating a hidden state update from $\mathcal{O}(d_\omega d_h^2)$ to $\mathcal{O}(d_\omega d_h^{1+\epsilon})$. Furthermore, it does not restrict the expressivity.

**Theorem 4.2.** *Sparse SLiCEs have maximal probabilistic expressivity.*

In theory, S-SLiCEs have faster training and inference times than DE-LNCDEs. In practice, current deep-learning frameworks (e.g. JAX [10], PyTorch [75]) are not optimised for unstructured sparsity, so practical speed-ups are not observed in our implementations. Nonetheless, we anticipate that ongoing work on sparse matrices will enable future gains in efficiency.

## 4.5 Walsh–Hadamard SLiCEs

A Hadamard matrix of order $n$ is an $n \times n$ matrix $H_n$ with entries $\pm 1$ whose rows (and columns) are mutually orthogonal, $H_n H_n^\top = n I_n$, where $I_n$ is the $n \times n$ identity matrix. When $n = 2^m$ for $m \in \mathbb{N}$, we can construct these matrices iteratively using the Sylvester construction [93]. Commonly, these matrices are applied via the Walsh–Hadamard transform (WHT), which admits an $\mathcal{O}(n \log n)$

algorithm [86]. Many scientific computing libraries include efficient CPU and GPU kernels for performing the Walsh–Hadamard transform [97, 1]. In practice, a normalisation factor of $n^{-1/2}$ can be applied to ensure the matrix is orthonormal [95].

Walsh–Hadamard SLiCEs (WH-SLiCEs) replace each dense matrix $A_\theta^i$ by the product

$$A_\theta^i = HD_\theta^i, \tag{12}$$

where $H$ is a normalised Hadamard matrix, and $D_\theta^i$ is a diagonal matrix. This parameterisation reduces the number of trainable parameters from $\mathcal{O}(d_\omega d_h^2)$ to $\mathcal{O}(d_\omega d_h)$. Summing the diagonal matrices across the channels and then applying the fast Walsh–Hadamard transform, the computational cost is $\mathcal{O}(\max(d_\omega d_h, d_h \log(d_h)))$. This is substantially cheaper than the $\mathcal{O}(d_\omega d_h^2)$ for dense LNCDEs. Furthermore, this modification does not restrict the expressivity.

**Theorem 4.3.** *Walsh–Hadamard SLiCEs have maximal probabilistic expressivity.*

### 4.6 Parallel computation

The recurrent cost of a SLiCE is based solely on the cost of calculating a single hidden state update, whereas the calculation when using an associative scan is repeatedly composing the flow

$$\exp\left(\sum_{i=1}^{d_\omega}(\omega_{t_{j+1}}^i - \omega_{t_j}^i)A_\theta^i\right) \approx I + \sum_{i=1}^{d_\omega}(\omega_{t_{j+1}}^i - \omega_{t_j}^i)A_\theta^i, \tag{13}$$

where the first-order approximation of the exponential is sometimes used in practice. When the $A_\theta^i$ are diagonal or block-diagonal, the composition of (13) preserves the structure, as these classes of matrices are closed under multiplication. Therefore, using a parallel associative scan reduces the scan depth from $n$ to $\log(n)$, whilst having a computational cost per composition of $\mathcal{O}(d_h)$ or $\mathcal{O}(d_h \sum_j b_j^2)$, respectively. However, for DPLR, sparse, and WH SLiCEs, the structured matrices are not closed under multiplication, which means that the limiting computational cost per composition is the same as a DE-LNCDE, $\mathcal{O}(d_h^3)$. Table 1 summarises the differences in parameter count, computational cost, existence of an efficient implementation, and expressivity of all the SLiCEs considered in this paper, where for simplicity we have taken $d_\omega = d_h$.

For large models, parallel associative scans result in high I/O costs, as each state-transition matrix must be materialised in GPU memory [100]. A possible approach to mitigating I/O costs for SLiCEs is combining them with the Log-ODE method. By approximating the solution over intervals, this method avoids explicitly materialising intermediate state-transition matrices. However, it does require computing the log-signature of the input path and iterated Lie brackets of the vector fields [98]. A detailed description of this approach is given in Appendix C and Table 1 quantifies the impact of the Log-ODE method on computational cost. In Section 5.3, we implement a hybrid strategy: the Log-ODE method is applied to small intervals, and the resulting outputs are then processed using a parallel associative scan. Yang et al. [100] introduced an alternative approach for DeltaNet, where a chunk-wise algorithm specifically tailored for diagonal-plus-rank-one state-transition matrices is used to bypass the need to materialise every intermediate matrix, significantly cutting down I/O costs [100]. Independently, Cirone and Salvi [18] and Siems et al. [88] extended this approach to higher rank matrices. These approaches can also be applied to diagonal state-transition matrices. Therefore, a block-diagonal SLiCE with a large diagonal portion ($b_i = 1$ for $i = 1, \ldots, k-1$) followed by a small dense block emerges as an attractive solution. The large diagonal section can efficiently utilise the chunk-wise algorithm and the smaller dense section can be processed using parallel associative scans without incurring significant I/O costs. We refer to this structure as diagonal-dense SLiCE (D-DE-SLiCE).

## 5 Experiments

### 5.1 The $A_5$ benchmark

The $A_5$ benchmark tests models on their ability to state-track [66]. Each sequence in the dataset consists of a series of permutations from the group of even permutations on five elements, denoted $A_5$. The task is to compose the permutations, which requires state-tracking. Following Merrill et al.

Table 1: **Comparison of SLiCEs on parameter count, computational cost, the existence of an efficient implementation, and expressivity.** Here, $d_h$ is the hidden dimension, $n$ is the sequence length, $b_j$ are BD's block-sizes, $r$ is DPLR's rank, $\epsilon$ is S's sparsity, and for simplicity we have taken $d_\omega = d_h$. Parallel cost is measured as $\mathcal{O}(\text{scan depth, cost per composition})$ when applying a parallel associative scan. Log-X-SLiCE corresponds to applying the Log-ODE method with fixed-size intervals containing $s$ samples and a truncation depth of $N$, where X is a specific SLiCE structure with $\mathcal{O}(P_X)$ parameters, $\mathcal{O}(R_X)$ recurrent cost, and $\mathcal{O}(C_X)$ cost per composition.

| Model | Parameters | Recurrent Cost | Parallel Cost | Efficient Impl. | Maximally Expressive |
|-------|-----------|----------------|---------------|-----------------|----------------------|
| DE-LNCDEs | $\mathcal{O}(d_h^3)$ | $\mathcal{O}(n d_h^3)$ | $\mathcal{O}(\log(n), d_h^3)$ | Yes | Yes |
| D-SLiCEs | $\mathcal{O}(d_h^2)$ | $\mathcal{O}(n d_h^2)$ | $\mathcal{O}(\log(n), d_h^2)$ | Yes | No |
| DPLR-SLiCEs | $\mathcal{O}(r d_h^2)$ | $\mathcal{O}(n r d_h^2)$ | $\mathcal{O}(\log(n), d_h^3)$ | Yes | Yes |
| S–SLiCEs | $\mathcal{O}(d_h^{2+\epsilon})$ | $\mathcal{O}(n d_h^{2+\epsilon})$ | $\mathcal{O}(\log(n), d_h^3)$ | No | Yes |
| WH–SLiCEs | $\mathcal{O}(d_h^2)$ | $\mathcal{O}(n d_h^2)$ | $\mathcal{O}(\log(n), d_h^3)$ | Yes | Yes |
| BD–SLiCEs | $\mathcal{O}\left(d_h \sum_j b_j^2\right)$ | $\mathcal{O}\left(n d_h \sum_j b_j^2\right)$ | $\mathcal{O}\left(\log(n), d_h \sum_j b_j^2\right)$ | Yes | Yes |
| Log–X-SLiCEs | $\mathcal{O}(P_X)$ | $\mathcal{O}\left(\frac{R_x}{s} d_h^{N-1}\right)$ | $\mathcal{O}\left(\log\left(\frac{n}{s}\right), C_X d_h^{N-1}\right)$ | - | - |

[66], we train and evaluate models on sequences ranging from length 3 to 20 and determine how many stacked layers each model needs to achieve a validation accuracy greater than 90%.

This benchmark serves as an empirical validation of our theoretical results; D-SLiCEs are less expressive than DPLR, sparse, WH, and BD SLiCEs. In addition to the SLiCEs, we consider Mamba [39], LSTM [44], gated DeltaProduct with negative eigenvalues [101, 88], and the two components of xLSTM [6] (mLSTM and sLSTM) on this benchmark. All baselines use a hidden dimension of 1024 and all SLiCEs use 1024 parameters per state-transition matrix. Full experimental details are given in Appendix D.1.

Figure 1a shows that the diagonal state-transition matrices of Mamba, mLSTM, and D-SLiCE mean that an increasing number of stacked layers are needed as the sequence length grows. Interestingly, Gated DeltaProduct with negative eigenvalues, which uses a DPLR structure, and the D-DE-SLiCE also need a growing number of stacked layers. However, DPLR and BD SLiCE both need one layer for all sequence lengths, suggesting this is not an inherent limitation of theses structures. Similarly, sparse, Walsh–Hadamard, and dense SLiCEs, as well as the two recurrent baselines LSTM and sLSTM, all need only one layer for all sequence lengths.

To assess length generalisation, we select the models that achieve at least 90% validation accuracy on sequences of length 20 and retrain them on sequences ranging from 3 to 40. Early stopping is performed using a validation set with sequence lengths from 40 to 128. The mLSTM is excluded because it requires fixed-length inputs. Figure 1b reports test accuracy for lengths from 20 to 5120. The recurrent LSTM and sLSTM generalise well, maintaining high test accuracy beyond both the training and validation ranges. Among the parallel-in-time models, three patterns emerge: (i) WH-SLiCE and Mamba do not attain high accuracy even at training lengths; (ii) DeltaProduct and D-DE-SLiCE generalise to approximately $2\times$ the training length but not beyond the validation range; and (iii) DE-LNCDE, DPLR-SLiCE, S-SLiCE, and BD-SLiCE sustain high accuracy on sequences at least $8\times$ the training length, exceeding the maximum validation length.

## 5.2 Regular language tasks

The formal language benchmark is a collection of language style tasks split into categories using the Chomsky hierarchy [17, 31]. Here, we use the regular tasks, which can be solved by processing inputs sequentially with a fixed set of internal states and no external memory, i.e. state-tracking. On this benchmark, the models are challenged to generalise to longer sequences, by training on sequences from length 3 to 40 and evaluating on sequences from length 40 to 256. Details on the individual tasks can be found in Appendix D.2. A wide range of existing sequence model architectures are used as baselines, including LSTM [44], xLSTM and its two components mLSTM and sLSTM [6], four variations of DeltaNet [85, 99, 101, 88, 38], RWKV-7 [76], a Transformer [96], S4D [40], and Mamba [39]. All models use two stacked layers. For each dataset and baseline model, we

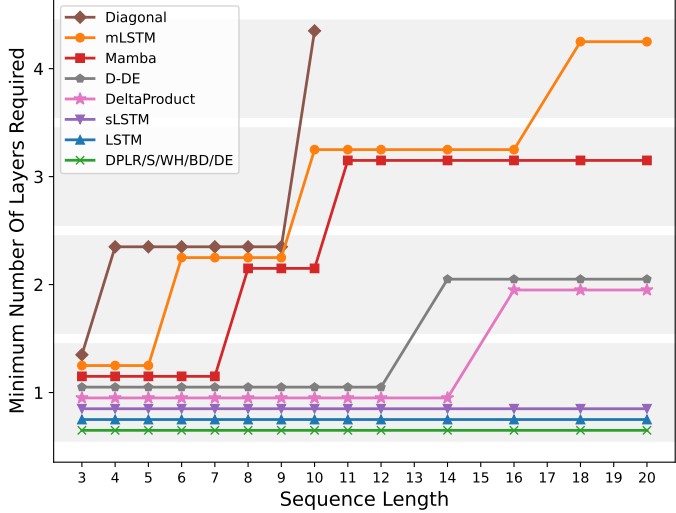

(a) Sequence length against the minimum number of stacked layers required to achieve greater than $90\%$ validation accuracy. Each shaded region indicates an equivalent number of stacked layers.

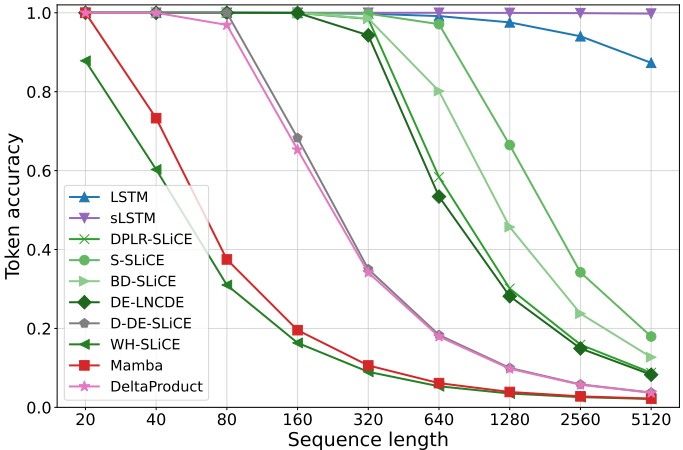

(b) Test set token accuracy (%) against sequence length.

Figure 1: **Results for $A_5$ Benchmark and $A_5$ length generalisation task.** Models evaluated are: Mamba, LSTM, mLSTM, sLSTM, Gated DeltaProduct with negative eigenvalues, and the SLiCEs.

selected the hidden dimension that yields a higher validation accuracy from two choices. The choices were 128 and 512 for all models aside from Mamba, and 256 and 512 for Mamba, as it does not support a hidden size of 128. All SLiCEs use two stacked layers and 512 non-zero parameters per state-transition matrix, except for the diagonal and Walsh–Hadamard, which also consider 128, with the better performing choice reported for each dataset. For the DPLR, block-diagonal, and diagonal-dense SLiCEs, we consider multiple choices of rank and block-size, respectively, and present the best performing models. A thorough investigation of the effect of block-size and rank is given in Appendix D.2. We do not consider S-SLiCE on this benchmark, due to the lack of an efficient implementation.

Table 2 presents the results. As expected, the recurrent LSTM generalises almost perfectly on all four tasks. Amongst the parallel models, DeltaNet with negative eigenvalues and Gated DeltaProduct with negative eigenvalues are the best performing baselines, aligning with the expectation that increased complexity in the state-transition matrix improves state-tracking performance. Similarly, D-SLiCE outperforms Mamba, aligning with the results of Grazzi et al. [38] that expanding the eigenvalue range of the state-transition matrix improves state-tracking performance. Similarly to the $A_5$ length

Table 2: **Results for formal language tasks.** Average and standard deviation of validation accuracy over five runs for a range of recurrent and parallel models.

| Model | Cycle Nav. | Even Pairs | Mod Arith. No Brack. | Parity | Average |
|---|---|---|---|---|---|
| **Recurrent** | | | | | |
| LSTM | $100.0 \pm 0.0$ | $100.0 \pm 0.0$ | $99.9 \pm 0.1$ | $100.0 \pm 0.0$ | 100 |
| sLSTM | $32.5 \pm 0.4$ | $100.0 \pm 0.0$ | $27.7 \pm 0.6$ | $100.0 \pm 0.0$ | 65.1 |
| xLSTM[1:1] | $53.5 \pm 5.6$ | $99.0 \pm 1.9$ | $29.3 \pm 1.6$ | $100.0 \pm 0.0$ | 70.5 |
| **Parallel** | | | | | |
| DeltaNet | $49.8 \pm 4.7$ | $100.0 \pm 0.0$ | $42.2 \pm 4.8$ | $57.8 \pm 0.8$ | 62.5 |
| DeltaNet$[-1, 1]$ | $46.7 \pm 6.1$ | $100.0 \pm 0.0$ | $66.4 \pm 8.8$ | $97.7 \pm 2.0$ | 77.7 |
| Gated DeltaNet | $53.8 \pm 8.8$ | $100.0 \pm 0.0$ | $42.8 \pm 8.2$ | $56.5 \pm 1.9$ | 63.3 |
| Gated DeltaProduct[-1,1] | $46.3 \pm 6.6$ | $100.0 \pm 0.0$ | $78.4 \pm 10.9$ | $98.0 \pm 1.4$ | 80.7 |
| RWKV-7 | $37.8 \pm 5.0$ | $88.1 \pm 14.2$ | $39.5 \pm 6.1$ | $51.1 \pm 0.3$ | 54.1 |
| mLSTM | $52.4 \pm 10.5$ | $99.9 \pm 0.1$ | $28.8 \pm 3.1$ | $53.0 \pm 2.1$ | 58.5 |
| Transformer | $24.4 \pm 0.5$ | $90.4 \pm 10.4$ | $23.6 \pm 0.7$ | $52.2 \pm 0.4$ | 47.7 |
| Mamba | $48.4 \pm 2.2$ | $100.0 \pm 0.0$ | $33.1 \pm 6.6$ | $54.2 \pm 2.1$ | 58.9 |
| S4D | $23.7 \pm 1.1$ | $68.7 \pm 4.7$ | $21.7 \pm 0.4$ | $51.2 \pm 1.0$ | 41.3 |
| D-SLiCE | $69.5 \pm 6.3$ | $100.0 \pm 0.0$ | $20.9 \pm 0.1$ | $100.0 \pm 0.0$ | 72.6 |
| WH-SLiCE | $69.7 \pm 8.8$ | $93.1 \pm 13.9$ | $23.8 \pm 1.1$ | $71.4 \pm 12.9$ | 64.5 |
| BD-SLiCE$_{d_h=128,b=4}$ | $99.8 \pm 0.2$ | $85.9 \pm 11.3$ | $54.0 \pm 12.5$ | $95.3 \pm 3.9$ | 83.8 |
| D-DE-SLiCE$_{d_h=272,b=16}$ | $73.3 \pm 29.4$ | $84.8 \pm 8.5$ | $98.4 \pm 0.7$ | $83.8 \pm 11.3$ | 85.1 |
| DPLR-SLiCE$_{d_h=57,r=4}$ | $81.1 \pm 16.6$ | $100.0 \pm 0.0$ | $68.3 \pm 19.3$ | $91.0 \pm 18.0$ | 85.1 |
| **Random** | 20.0 | 50.0 | 20.0 | 50.0 | 35.0 |

generalisation task, the WH-SLiCE underperforms other SLiCE structures. However, unlike in the $A_5$ length generalisation task, the D-DE-SLiCE achieves the joint highest average validation accuracy among the parallelisable models, alongside DPLR-SLiCE.

## 5.3 UEA multivariate time-series classification archive

Since SLiCEs are descendants of NCDEs, they inherit a number of the desirable properties which arise from having a natural continuous-time formulation. These include robustness to irregular sampling rates and decoupling the number of recurrent steps from the number of observations in the time-series [53, 98]. Furthermore, SLiCEs have the same theoretical expressivity as NCDEs, whilst being parallel-in-time, making them an attractive alternative for real-world time-series modelling.

As a demonstration of the practical benefits, we consider six datasets from the UEA Multivariate Time-Series Classification Archive (UEA-MTSCA), a collection of time-series classification tasks, ranging from classifying worms into species based on movement to classifying alcohol by concentration using vibrational spectroscopy [4]. Walker et al. [98] showed that Log-NCDEs outperform the linear recurrent unit (LRU) [73], S5 [89], S6 [39], and Mamba [39] on average test accuracy over the six longest datasets with at least 200 observations [98]. However, the per-step training time is significantly higher for Log-NCDEs than the baseline methods.

Keeping all other hyperparameters the same, Table 3 presents the impact of replacing the non-linear vector field of a Log-NCDE with a structured linear vector field. The GPU memory and time per 1000 training steps were recalculated for all models on an NVIDIA H100. BD-SLiCE achieves similar performance to the Log-NCDE, whilst reducing the average per-step training time by a factor of nearly 20 and increasing the average GPU memory usage by only $8\%$. Appendix D.3 presents the results for individual datasets and analyses the impact of the Log-ODE method and parallel associative scan on run-time and GPU memory.

## 6 Limitations and Future Work

To reduce the computational burden of SLiCEs, our implementation approximates the matrix exponential when computing the flow via (13). However, even with this adjustment, scaling SLiCEs to

Table 3: **Average test accuracy, rank, training time per 1,000 steps, and GPU memory usage across six datasets from the UEA-MTSCA.** All SLiCE variants use a parallel associative scan during training. Therefore, the Walsh–Hadamard, DPLR, and sparse SLiCEs are treated as dense LNCDEs (see Section 4.6), and their timing and GPU memory results are omitted.

| Model | Av. Acc | Av. Rank | Av. Time / 1k Steps (s) | Av. GPU mem (MB) |
|---|---|---|---|---|
| BD-SLiCE | 64.0 | 3.2 | 68.1 | 2344 |
| Log-NCDE | 64.3 | 4.0 | 1321.7 | 2177 |
| D-DE-SLiCE | 63.0 | 5.7 | 66.7 | 2302 |
| WH-SLiCE | 62.5 | 6.7 | – | – |
| DPLR-SLiCE | 62.0 | 7.0 | – | – |
| D-SLiCE | 61.7 | 7.2 | 11.0 | 1875 |
| LRU | 61.7 | 7.3 | 26.9 | 4308 |
| S6 | 62.0 | 7.7 | 20.1 | 2938 |
| S5 | 61.8 | 8.0 | 21.9 | 3327 |
| S-SLiCE | 61.8 | 8.2 | – | – |
| DE-LNCDE | 61.6 | 8.3 | 77.2 | 12756 |
| NCDE | 60.2 | 8.8 | 6923 | 1962 |
| NRDE | 60.6 | 10.3 | 3431 | 2858 |
| Mamba | 58.6 | 10.8 | 60.0 | 4535 |

the multi-billion parameter regime remains challenging. A key technical goal is the development of efficient GPU kernels for the matrix exponential and parallel associative scans, particularly when handling many small independent systems, such as for BD-SLiCEs. Alternatively, building on the work of Yang et al. [100] and Cirone and Salvi [18], fast chunk-wise methods for a broader class of structured matrices may offer a viable path forward.

Alternative SLiCE architectures may achieve maximal expressivity and improved empirical performance. A theoretical characterisation of the conditions that a SLiCE's structured matrix needs to satisfy to achieve maximal probabilistic expressivity would aid the search for additional structures. Moreover, although establishing maximal probabilistic expressivity is a significant step towards a deeper theoretical understanding of structured state-transition matrices, expressivity at finite hidden dimensions remains an open challenge.

Finally, unlike NCDEs, SLiCEs are sequence-to-sequence models that update their state with each input sample. Therefore, similarly to other discrete sequence models, SLiCEs are susceptible to over-sampled data. Combining SLiCEs with the Log-ODE method enables path-based inputs by operating with flows over intervals, rather than individual samples. However, the Log-SLiCE outputs a sequence whose elements correspond to the boundary values of the output path for each interval the Log-ODE method was applied to. Therefore, you cannot stack two Log-SLiCEs, as the first level has produced a sequence, whereas the second level consumes a path. A natural direction for future work is developing a true path-to-path model.

## 7   Conclusion

This paper introduced SLiCEs, a unifying framework for sequence-to-sequence layers that are maximally expressive, computationally efficient, and allow for parallel-in-time computation. We explored four specific instances, diagonal-plus-low-rank, sparse, Walsh–Hadamard, and block-diagonal, analysing their theoretical properties and empirical performance. Theorems 4.1, 4.2, and 4.3 established that block-diagonal, sparse, and Walsh–Hadamard SLiCEs achieve maximal probabilistic expressivity. Furthermore, all SLiCE structures demonstrated single-layer state-tracking on the $A_5$ benchmark, unlike the other parallelisable layers considered: diagonal SLiCEs, mLSTM, Mamba, and DeltaProduct. Among the SLiCEs, block-diagonal stands out as the only maximally expressive variant that strictly reduces parameter count, recurrent cost, and parallel cost compared to dense LNCDEs. Additionally, a variant of the block-diagonal SLiCE achieved the joint highest average validation accuracy among parallel models on the regular language tasks from the formal language benchmark. Finally, practical speed-ups for real-world time series modelling were demonstrated on six multivariate time-series classification datasets, where replacing the non-linear vector field of a Log-NCDE with a block-diagonal linear vector field reduced the average time per training step by a factor of twenty, without impacting the model's overall performance.

## Acknowledgements

We thank Joël Mouterde, Jérôme Tomezyk, Sam Morley, and Alexandre Bloch for engaging and insightful discussions on the design and training of linear neural controlled differential equations. We thank Merrill et al. [66], Delétang et al. [31], and Bagnall et al. [4] for the $A_5$, formal language, and UEA datasets respectively.

Benjamin Walker was funded by the Hong Kong Innovation and Technology Commission (InnoHK Project CIMDA). Lingyi Yang is supported by EPSRC [EP/S026347/1] and the Hong Kong Innovation and Technology Commission (InnoHK Project CIMDA). Nicola Muca Cirone is supported by the EPSRC Centre for Doctoral Training in Mathematics of Random Systems: Analysis, Modelling and Simulation [EP/S023925/1]. Terry Lyons was supported in part by the UKRI EPSRC through the Programme Grants High order mathematical and computational infrastructure for streamed data that enhance contemporary generative and large language models (UKRI1010) and Unparameterised multi-model data, high order signatures and the mathematics of data science (EP/S026347/1) and UKRI AI for Science award UKRI2385; he was supported by The Alan Turing Institute under the EPSRC Grant EP/N510129/1, the Defence and Security Programme (funded by the UK Government), and through CIMDA@Oxford, part of the AIR@InnoHK initiative funded by the Innovation and Technology Commission, HKSAR Government. The authors would like to acknowledge the use of the University of Oxford Advanced Research Computing (ARC) facility in carrying out this work: http://dx.doi.org/10.5281/zenodo.22558. For the purpose of Open Access, the author has applied a CC BY public copyright licence to any Author Accepted Manuscript (AAM) version arising from this submission.

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

# A Linear Neural Controlled Differential Equations

## A.1 The Connection to Structured State-Space Models

Viewing SSMs as LNCDEs provides a theoretical framework for comparing architectures and reasoning about their expressivity. Following the approach of Cirone et al. [22], this appendix shows how S6, the recurrent core of Mamba, can be recast as an LNCDE. Additionally, this chapter highlights the limits of diagonal state-transition matrices, details the connection between our work and matrix-valued hidden states, and provides further implementation details for our SLiCE models.

S6 is defined by

$$h_{t_{i+1}}^j = \bar{C}_\theta^j(x_{t_i})h_{t_i}^j + \bar{D}_\theta^j(x_{t_i})x_{t_i}^j, \tag{14}$$

where $j$ denotes the input channel,

$$
\begin{aligned}
\bar{C}_\theta^j(x_{t_i}) &= \exp(\Delta^j(x_{t_i})C_\theta), \\
\bar{D}_\theta^j(x_{t_i}) &= (\Delta^j(x_{t_i})C_\theta)^{-1}(\exp(\Delta^j(x_{t_i})C_\theta) - I)\Delta^j(x_{t_i})D_\theta x_{t_i}, \\
&\approx \Delta^j(x_{t_i})D_\theta x_{t_i},
\end{aligned}
\tag{15}
$$

with trainable parameters $D_\theta \in \mathbb{R}^{d_h \times d_x}$ and a diagonal state-transition matrix $C_\theta \in \mathbb{R}^{d_h \times d_h}$, and

$$\Delta^j(x_{t_i}) = \text{softplus}(\alpha_\theta^j \cdot x_{t_i} + \beta_\theta^j), \tag{16}$$

with trainable parameters $\alpha_\theta^j \in \mathbb{R}^{d_x}$ and $\beta_\theta^j \in \mathbb{R}$ [39]. Equation (14) can be considered a zero-order hold discretisation of

$$\mathrm{d}h_s^j = C_\theta \Delta^j(X_s)h_s^j + D_\theta X_s \Delta^j(X_s)X_s^j \mathrm{d}s, \tag{17}$$

where $X_s$ is an interpolation of $\{x_{t_i}\}_{i=0}^n$. As shown by Cirone et al. [22], these equations can be stacked and rewritten as an affine LNCDE,

$$h_t = h_{t_0} + \int_{t_0}^t \sum_{i=0}^{d_x} A_\theta^i h_s \mathrm{d}\omega_s^{X,i} + \int_{t_0}^t B_\theta \mathrm{d}\xi_s^X, \tag{18}$$

where

$$
\begin{aligned}
\omega_t^{X,i} &= \int_{t_0}^t \text{softplus}(\alpha^i \cdot X_s + \beta^i)\mathrm{d}s, \\
\xi_t^X &= \int_{t_0}^t \begin{bmatrix} X_t\text{softplus}(\alpha^1 \cdot X_s + \beta^1)X_s^1 \\ \vdots \\ X_t\text{softplus}(\alpha^{d_x} \cdot X_s + \beta^{d_x})X_s^{d_x} \end{bmatrix} \mathrm{d}s,
\end{aligned}
\tag{19}
$$

and

$$
\begin{aligned}
A_\theta^i &= \text{diag}(0,\dots,0,C_\theta,0,\dots,0) \in \mathbb{R}^{d_h d_x \times d_h d_x}, \\
B_\theta &= \text{diag}(D_\theta,\dots,D_\theta) \in \mathbb{R}^{d_h d_x \times d_x^2},
\end{aligned}
\tag{20}
$$

with the non-zero diagonal element of $A_\theta^i$ in the $i^{\text{th}}$ position. Neglecting the bias term gives a specific instance of an LNCDE,

$$h_t = h_{t_0} + \int_{t_0}^t \sum_{i=0}^{d_x} A_\theta^i h_s \mathrm{d}\omega_s^{X,i}. \tag{21}$$

The two core differences between the LNCDEs used in this paper and Mamba are:

1. More general forms for the driving path $\omega_t^X$. For example, the LNCDEs in this paper use

$$\frac{\omega_{t_{j+1}}^X - \omega_{t_j}^X}{t_{j+1} - t_j} = (1, X_{t_j}) \tag{22}$$

   on the $A_5$ and formal language benchmarks and $\omega_t^X = X_t$ on the UEA benchmarks.

2. Mamba uses diagonal $A_\theta^i$, whereas an LNCDE uses dense $A_\theta^i$.

Cirone et al. [22] use the LNCDE perspective to provide a full theoretical analysis on the impact of using diagonal matrices instead of dense matrices. Here, we give a simplistic example demonstrating the difference in expressivity between diagonal and dense state-transition matrices.

Consider a stream of bits

$$x_1, x_2, \ldots, x_n \in \{0, 1\}, \tag{23}$$

where we want to predict the parity label defined by

$$p_n = S_n \bmod 2 \in \{0, 1\}, \quad S_n = \sum_{k=1}^{n} x_k. \tag{24}$$

Whenever a new bit is 1 the label flips; if the bit is 0 the label stays the same. Taking a diagonal LNCDE with a hidden dimension of 2 and $\omega_{k+1}^x - \omega_k^x = x_{k+1}$, then

$$h_{n+1} = \exp\left( \begin{bmatrix} a_1 & 0 \\ 0 & a_2 \end{bmatrix} x_{n+1} \right) h_n, \tag{25}$$

and

$$h_n^i = h_0^i \exp(a_i S_n), \qquad i = 1, 2. \tag{26}$$

With a linear read-out $r = (r_1, r_2)^\top$ followed by a monotone activation $\phi$ (such as tanh, ReLU, sigmoid):

$$\hat{p}_n = \phi\left( r^\top h_n \right) = \phi\left( c_1 e^{a_1 S_n} + c_2 e^{a_2 S_n} \right) = \phi\left( f(S_n) \right), \qquad c_i = r_i h_0^i. \tag{27}$$

Since $f(S)$ has at most one turning point, and $\phi$ is monotone, $\hat{p}_n$ can cross any chosen threshold at most twice. However, the true label $p_n$ flips every time $S_n \mapsto S_n + 1$. Hence, no diagonal $2 \times 2$ LNCDE can realise parity on arbitrarily long input. Similarly, for a hidden dimension of $n$, $f(S)$ can have at most $n - 1$ turning points, so no diagonal LNCDE with a fixed hidden dimension can realise parity on arbitrarily long input. If you replace $A$ with

$$A = \begin{pmatrix} 0 & \pi \\ -\pi & 0 \end{pmatrix}. \tag{28}$$

then

$$\exp(A x_{n+1}) = \begin{pmatrix} 1 & 0 \\ 0 & 1 \end{pmatrix}, \tag{29}$$

when $x_{n+1} = 0$ and

$$\exp(A x_{n+1}) = \begin{pmatrix} -1 & 0 \\ 0 & -1 \end{pmatrix}, \tag{30}$$

when $x_{n+1} = 1$. Thus

$$h_{n+1} = (-1)^{x_{n+1}} h_n \tag{31}$$

and

$$h_n = (-1)^{S_n} h_0. \tag{32}$$

Taking $r = (1, 0)^\top$, $h_0^{(1)} = 1$, and $\phi(s) = \left( 1 - \text{sign}(s) \right)/2$,

$$\hat{p}_n = \frac{1 - \text{sign}\left( (-1)^{S_n} \right)}{2} = S_n \bmod 2 = p_n. \tag{33}$$

Therefore, a dense LNCDE can solve parity exactly with a hidden dimension of 2.

This example aligns with the results of Cirone et al. [22]: diagonal $A_\theta^i$ are fundamentally less expressive than dense $A_\theta^i$. However, as noted in Section 2.2, the additional computational cost makes dense state-transition matrices infeasible in practice. This trade-off between efficiency and expressivity is the motivation for the structured $A_\theta^i$ in the SLiCEs introduced in this paper.

## A.2 Matrix-valued Hidden States

When rewriting Mamba as an affine LNCDE (18), the hidden states for each channel of the input are stacked vertically. An alternative approach is to view the hidden states for each channel as the columns of a matrix, producing a LRNN with a matrix-valued hidden state.

Matrix-valued LRNNs originated as an alternative viewpoint on linear Transformers, where softmax is replaced with a kernel admitting a finite-dimensional feature map, $\kappa(q, k) = \phi(q)^\top \phi(k)$ for $\phi : \mathbb{R}^{d_q} \to \mathbb{R}^{d_\phi}$. Letting $W_\theta^q \in \mathbb{R}^{d_q \times d_x}$, $W_\theta^k \in \mathbb{R}^{d_q \times d_x}$, and $W_\theta^v \in \mathbb{R}^{d_v \times d_x}$ be learnable weights, then causal linear attention is defined by

$$o_{t_i} = \frac{\sum_{j=1}^i \phi\big(W_\theta^q x_{t_i}\big)^\top \phi\big(W_\theta^k x_{t_j}\big) W_\theta^v x_{t_j}}{\sum_{j=1}^i \phi\big(W_\theta^q x_{t_i}\big)^\top \phi\big(W_\theta^k x_{t_j}\big)}. \tag{34}$$

As shown by Katharopoulos et al. [50], this admits an RNN formulation,

$$\begin{aligned}
S_{t_i} &= S_{t_{i-1}} + \phi\big(W_\theta^k x_{t_i}\big)\big(W_\theta^v x_{t_i}\big)^\top, \\
z_{t_i} &= z_{t_{i-1}} + \phi\big(W_\theta^k x_{t_i}\big), \\
o_{t_i} &= \frac{\phi\big(W_\theta^q x_{t_i}\big)^\top S_{t_i}}{\phi\big(W_\theta^q x_{t_i}\big)^\top z_{t_i}},
\end{aligned} \tag{35}$$

where $S_i \in \mathbb{R}^{d_\phi \times d_v}$ with $S_0 = 0$, $z_i \in \mathbb{R}^{d_\phi}$ with $z_0 = 0$, and $o_i \in \mathbb{R}^{d_v}$. Ignoring the normalisation, the core recurrence is a matrix-valued LRNN.

Matrix-valued LRNNs are a useful framework for understanding several recent sequence models, as highlighted by Yang et al. [100]. Many of these models share the general form:

$$S_{t_{i+1}} = S_{t_i} \bullet M_{t_i} + v(x_{t_i})k(x_{t_i})^\top, \tag{36}$$

where $\bullet$ denotes an associative operator and $v$ and $k$ are potentially non-linear functions of $x_t$. For example, Mamba's recurrence (14) can be rewritten as

$$S_{t_{i+1}} = S_{t_i} \odot M_{t_i} + v(x_{t_i})k(x_{t_i})^\top, \tag{37}$$

where $\odot$ refers to the Hadamard (element-wise) product and $S \in \mathbb{R}^{d_h \times d_x}$ [100]. The Hadamard product is a direct consequence of Mamba's diagonal state-transition matrix, which inherently prevents interaction between individual elements of the hidden state. This framework gives a clear interpretation for a key modification to the recurrence introduced by Mamba-2 [28],

$$S_{t_{i+1}} = \gamma(x_{t_i})S_{t_i} + v(x_{t_i})k(x_{t_i})^\top, \tag{38}$$

where $\gamma$ is a real-valued function. Additionally, it highlights the structural similarity between Mamba and linear Transformers.

Replacing the Hadamard product by a matrix product allows modelling richer interactions between the hidden states,

$$S_{t_{i+1}} = S_{t_i} M_{t_i} + v(x_{t_i})k(x_{t_i})^\top. \tag{39}$$

However, similarly to using dense matrices in an LNCDE, the cost makes models of this type intractable in larger models. DeltaNet uses a diagonal-plus-rank-one structure,

$$S_{t_{i+1}} = S_{t_i}(I - \beta(x_{t_i})k(x_{t_i})k(x_{t_i})^\top) + \beta(x_{t_i})v(x_{t_i})k(x_{t_i})^\top, \tag{40}$$

where $\beta$ is a real-valued function [85, 100]. DeltaProduct later generalised DeltaNet to DPLR matrices [88]. Beyond reducing parameter count and recurrent computational cost, the DPLR structure of DeltaNet also facilitates an efficient chunk-wise algorithm that can outperform parallel associative scans for large hidden dimensions, as outlined in [100, Section 3.2].

Many recent sequence models can also be viewed as matrix-valued LRNNs. These include, but are not limited to, Gated DeltaNet [101], RWKV-7 [76], HGRN-2 [79], mLSTM [6], Gated Linear Attention [99], Gated Random Feature Attention [77], Gated Slot Attention [102], TTT-Linear [92], and Titans [7]. A detailed comparison of the specific form of (36) for many of these models is provided in Table 2 of [100]. Beck et al. [6] introduced mLSTM alongside a non-linear recurrent

model sLSTM, which together form their sequence model xLSTM. These components are designed to play different roles, with mLSTM acting as the memory and sLSTM performing the reasoning. Section 5 uses mLSTM, sLSTM, and xLSTM as baseline methods to highlight the different roles the components play.

Matrix-valued LRNNs can be converted into vector-valued LRNNs by returning to the stacking approach of Section A.1. Let $\text{vec} : \mathbb{R}^{m \times n} \to \mathbb{R}^{mn}$ be the column-major vectorisation operator, which transforms a matrix into a vector by stacking its columns. Letting $\otimes$ denote the Kronecker product, the Hadamard product LRNN (37) can be rewritten as

$$\text{vec}(S_{t_{i+1}}) = \text{diag}(\text{vec}(M_{t_i})) \, \text{vec}(S_{t_i}) + k(x_{t_i}) \otimes v(x_{t_i}), \tag{41}$$

where the state-transition matrix is diagonal, consistent with Mamba's design. Noting that

$$\text{vec}(AXB) = (B^\top \otimes A) \, \text{vec}(X), \tag{42}$$

then the LRNN with a matrix product (39) can be written as

$$\text{vec}(S_{t_{i+1}}) = (M_{t_i}^\top \otimes I_{d_h}) \, \text{vec}(S_{t_i}) + k(x_{t_i}) \otimes v(x_{t_i}). \tag{43}$$

In both cases, a matrix-valued recurrence is equivalent to a vector-valued recurrence on a flattened hidden state, where the state-transition matrix is constrained to have a specific structure. For the Hadamard product, this gives a diagonal matrix, while the matrix product leads to a Kronecker product structure.

This work is focused on the vector-valued case. The insights gained from this analysis naturally extend to the matrix-valued setting. For instance, the limitations in expressivity of diagonal matrices directly translate to the Hadamard product formulation. Similarly, the Kronecker product structure arising from vectorising a matrix-valued recurrence clearly illustrates how the properties of the constituent matrices determine the properties of the overall state transition.

LNCDEs can be generalised to matrix-valued hidden states by reversing the above vectorisation. Alternatively, one can run $m$ copies of the same LNCDE via

$$H_t = H_{t_0} + \int_{t_0}^t \left( \sum_{i=1}^{d_\omega} A_\theta^i \, \mathrm{d}\omega_s^{X,i} \right) H_s, \tag{44}$$

where $H_s \in \mathbb{R}^{d_h \times m}$. Clearly, $m$ copies of a vector-valued LNCDE or SLiCE match the expressivity of a single copy, so all of our theoretical results naturally carry over to this setting. However, the columns only differ due to their initial conditions. To introduce meaningful column-specific dynamics, you can include a column-specific bias term in the vector field. A natural way to incorporate such biases into a matrix-valued LNCDE is to consider an affine LNCDE,

$$H_t = H_{t_0} + \int_{t_0}^t \left( \sum_{i=1}^{d_\omega} A_\theta^i \, \mathrm{d}\omega_s^{X,i} \right) H_s + B_\theta \, \text{diag}(\mathrm{d}\xi_s^X), \tag{45}$$

where $B_\theta \in \mathbb{R}^{d_h \times m}$ and $\xi^X : [0, T] \to \mathbb{R}^m$ is a path, dependent on the input $X$. Letting $h_t^k$ for $1 \leq k \leq m$ denote the columns of $H_t$, then

$$h_t^k = h_{t_0}^k + \int_{t_0}^t \left( \sum_{i=1}^{d_\omega} A_\theta^i \, \mathrm{d}\omega_s^{X,i} \right) h_s^k + B_\theta^k \mathrm{d}\xi_s^{X,k}. \tag{46}$$

Approximating $\omega_s$ and $\xi_s$ with linear interpolation on the grid $0 = t_0 < \cdots < t_n = T$ yields

$$\tilde{h}_{t_{j+1}}^k \approx \exp\left( \sum_{i=1}^{d_\omega} A_\theta^i \left( \omega_{t_{j+1}}^{X,i} - \omega_{t_j}^{X,i} \right) \right) \tilde{h}_{t_j}^k + B_\theta^k (\xi_{t_{j+1}}^{X,k} - \xi_{t_j}^{X,k}), \tag{47}$$

where we have used the approximation

$$\int_0^{t_{j+1} - t_j} \exp\left( \tau \sum_{i=1}^{d_\omega} A_\theta^i \frac{\omega_{t_{j+1}}^{X,i} - \omega_{t_j}^{X,i}}{t_{j+1} - t_j} \right) \mathrm{d}\tau \approx (t_{j+1} - t_j)I, \tag{48}$$

for small increments. The outputs $\tilde{h}_{t_j}^k$ remain computable in $\mathcal{O}(\log(n))$ parallel steps using a parallel associative scan [9].

### A.3 Implementation Details

Algorithm 1 provides a pseudo-code implementation for the forward pass of a SLiCE. The approach is demonstrated for dense state-transition matrices $A_\theta^i$, with comments highlighting where a SLiCE's structure can be used to speed up computation or reduce memory footprint. In this paper, each SLiCE recurrence is embedded in a simple block structure, combining a linear layer to mix the channels, tanh activation function, layer normalisation, and a skip connection. Inspired by the $\mathrm{Lip}(\gamma)$ regularisation of Log-NCDEs, all SLiCEs employ weight regularisation on their state transition matrices.

---

**Algorithm 1 Structured Linear CDE**: The algorithm is presented for a dense state-transition matrix $A_\theta$. Comments indicate where the structure of $A_\theta$ can be used to reduce memory footprint and speed-up computation.

---

**Input:** $\boldsymbol{\omega} : (B, L, d_\omega)$
**Output:** $\mathbf{h} : (B, L, d_h)$
 1: $\mathbf{h}_0 : (B, d_h) \leftarrow \xi_\phi(\boldsymbol{\omega}_0)$
 2: $\boldsymbol{\omega}^{\mathrm{inc}} : (B, L-1, d_\omega) \leftarrow \mathrm{diff}(\boldsymbol{\omega})$
 3: $A_\theta : (d_\omega, d_h, d_h) \leftarrow$ *Parameter*              $\triangleright$ Exploit structure of $A_\theta$
 4: $\mathbf{I} : (d_h, d_h) \leftarrow d_h \times d_h$ identity matrix.
 5: **if** *mode* = parallel **then**
 6:   $\mathbf{F} : (B, L-1, d_h, d_h) \leftarrow \mathbf{I} + \mathrm{einsum}(bli, ijk \to bljk, \boldsymbol{\omega}^{\mathrm{inc}}, A_\theta)$  $\triangleright$ Broadcast $\mathbf{I}$, exploit structure of $A_\theta$
 7:   $\mathbf{F}^{\mathrm{comp}} \leftarrow \mathrm{pscan}(\mathbf{F})$          $\triangleright$ Parallel associative scan, exploit structure of $\mathbf{F}$
 8:   $\mathbf{h}_{1:L-1} \leftarrow \mathrm{einsum}(bljk, bk \to blj, \mathbf{F}^{\mathrm{comp}}, \mathbf{h}_0)$
 9:   $\mathbf{h} \leftarrow [\mathbf{h}_0, \mathbf{h}_{1:L-1}]$
10: **else**                         $\triangleright$ Recurrent pass
11:   **for** $t \leftarrow 0$ **to** $L-2$ **do**
12:     $\mathbf{F}_t : (B, d_h, d_h) \leftarrow \mathbf{I} + \mathrm{einsum}(bi, ijk \to bjk, \boldsymbol{\omega}_t^{\mathrm{inc}}, A_\theta)$  $\triangleright$ Exploit structure of $A_\theta$
13:     $\mathbf{h}_{t+1} \leftarrow \mathrm{einsum}(bjk, bk \to bj, \mathbf{F}_t, \mathbf{h}_t)$      $\triangleright$ Exploit structure of $\mathbf{F}$
14:   **end for**
15:   $\mathbf{h} \leftarrow [\mathbf{h}_0, \dots, \mathbf{h}_{L-1}]$            $\triangleright$ Stack along length axis
16: **end if**
17: **return** $\mathbf{h}$

---

## B Expressivity

### B.1 Introduction

Rough path theory [13, 61] provides a mathematical framework for analysing continuous-time paths $X : [0, T] \to \mathbb{R}^d$. Because it is formulated in continuous time, it can be applied to time series observed at irregular sampling times, a common situation in real-world applications. A central object is the path signature, defined in (51), which is a graded sequence of iterated integrals that characterises the path. The signature encodes geometric features such as increments and areas, it determines the path up to tree-like equivalence [42], and linear maps of signatures are maximally expressive on suitable compact sets of paths in the sense of Definition 3.1. The combination of maximal expressivity and a natural grading makes signature coefficients particularly well-suited as feature representations for machine learning tasks involving sequential data [58, 34, 11]. These techniques have experienced substantial growth in popularity and have been successfully implemented across diverse domains, with applications spanning deep learning [37, 52, 70, 71, 20, 45, 22, 98, 49, 5, 78, 8], kernel methods [55, 82, 81, 57, 56, 63], and quantitative finance [2, 83, 48, 24, 74, 19]. Additionally, signature methods have proven valuable in information theory [84, 87], cybersecurity [23], sepsis detection [69, 25], and computational neuroscience [46], demonstrating their versatility across scientific disciplines. Signature methods are practical thanks to efficient, well-developed packages for computing them [32, 68]. Some introductory texts to signatures and rough path theory are [15] and [62]. This section utilises the tools of rough path theory to characterise the expressivity of the structured linear controlled differential equations introduced in this paper.

The core of our models is the linear controlled differential equation (1):

$$\mathrm{d}h_s = \sum_{i=1}^{d_\omega} A^i h_s \mathrm{d}\omega_s^i, \quad h_0 \in \mathbb{R}^{d_h}. \tag{49}$$

By [22, Proposition B.4], this can be written in terms of the signature as

$$\mathbf{h}((A^i)_i, h_0, \omega)_t := h_t = \sum_{I \in \mathbb{W}_{d_\omega}} (A^I h_0) \, S^I(\omega)_{[0,t]}, \tag{50}$$

where $\mathbb{W}_{d_\omega}$ is the set of words in the alphabet $[[d_\omega]] := \{1, \ldots, d_\omega\}$ (i.e. $\mathbb{W}_{d_\omega} = \bigcup_{n \geq 0} [[d_\omega]]^n$ ) and for a given word $I = i_1 \ldots i_n$, $S^I(\omega)_{[0,t]}$ referrs to the $I$th component of the signature tensor $S(\omega)_{[0,t]}$,

$$S^I(\omega)_{[0,t]} = \underbrace{\int \cdots \int}_{\substack{u_1 < \cdots < u_n \\ u_i \in [0,t]}} \mathrm{d}\omega_{u_1}^{i_1} \cdots \mathrm{d}\omega_{u_n}^{i_n}. \tag{51}$$

It is evident from (50) that any linear readout of $h_t$ is expressed as a series in signature terms. Consequently, such systems are inherently limited to learning functions that are close to these (uniformly convergent) series. Maximal expressivity is thus achieved when any finite linear combination in signature terms can be approximated by a linear readout on $h_t$ through appropriate choices of the matrices $A^i$.

**Definition B.1.** *Fix a set of paths $\mathcal{X} \subseteq C^{1-var}([0,1]; \mathbb{R}^d)$. We say that a sequence $(\mathcal{A}_N, \mathcal{H}_N)_{N \in \mathbb{N}}$, where $\mathcal{H}_N \subseteq \mathbb{R}^N$ and $\mathcal{A}_N \subseteq \mathbb{R}^{N \times N}$, achieves maximal expressivity for $\mathcal{X}$ whenever for any positive tolerance $\epsilon > 0$ and any finite linear combination coefficients $\alpha \in T(\mathbb{R}^d)$, there exists a choice of parameters $v, (A^i), h_0$ in some $\mathbb{R}^N, \mathcal{A}_N, \mathcal{H}_N$ in the sequence, such that $v^\top \mathbf{h}((A^i), h_0, \omega)$ is uniformly close to $\langle \alpha, S(\omega)_{[0,\cdot]} \rangle$ up to an error of $\epsilon$,*

$$\forall \epsilon > 0, \ \forall \alpha \in T(\mathbb{R}^d), \ \exists N \geq 0, \ \exists (v, (A^i), h_0) \in \mathbb{R}^N \times \mathcal{A}_N^d \times \mathcal{H}_N \text{ s.t.}$$

$$\sup_{(\omega,t) \in \mathcal{X} \times [0,1]} |\langle \alpha, S(\omega)_{[0,t]} \rangle - v^\top \mathbf{h}((A^i), h_0, \omega)_t| < \epsilon.$$

*If we are given a sequence of probabilities $\mathbb{P}_N$ on $\mathcal{A}_N^d \times \mathcal{H}_N$ such that $\forall \epsilon > 0$, $\forall \alpha \in T(\mathbb{R}^d)$, it holds that*

$$\lim_{N \to \infty} \mathbb{P}_N \left\{ \exists v \in \mathbb{R}^N \text{ s.t.} \sup_{(\omega,t) \in \mathcal{X} \times [0,1]} |\langle \alpha, S(\omega)_{[0,t]} \rangle - v^\top \mathbf{h}((A^i), h_0, \omega)_t| < \epsilon \right\} = 1, \tag{52}$$

*then we say that $(\mathcal{A}_N, \mathcal{H}_N, \mathbb{P}_N)_{N \in \mathbb{N}}$ achieves maximal probabilistic expressivity for $\mathcal{X}$.*

A deterministic argument by [51] demonstrates the existence of a specific choice of $\mathcal{A}_N, \mathcal{H}_N$ that mimics the algebraic structure of tensors and provides maximal expressivity for compact sets of paths. Furthermore, Cirone et al. [22, Theorem B.13] established that matrices (almost) replicating the algebraic structure of tensors are, in fact, abundant. They showed that the triplet $(\mathbb{R}^{N \times N}, \mathbb{R}^N, \mathbb{P}_N)$, where $\mathbb{P}_N$ is a Gaussian measure achieves maximal probabilistic expressivity for compact sets.

The result in [22] implies that for dense matrices $A^i$, if the hidden dimension $N$ is sufficiently large, there is a significant abundance of parameters capable of achieving uniformly and arbitrarily low error rates. These parameters should therefore be readily discoverable through standard optimisation methods. Unfortunately, as discussed in Section 2.2, using dense matrices is infeasible in practice due to computational constraints. In this section, we present three alternative choices of parameters that lead to maximal probabilistic expressivity for compact sets. These alternatives offer better computational properties compared to the naive use of dense matrices.

## B.2 Sparse Matrices

**Proposition B.2.** *The sequence of triplets $(\mathbb{R}^{N \times N}, \mathbb{R}^N, \mathbb{P}_N)$ where $\mathbb{P}_N$ is such that*

- *the initial value has independent standard Gaussian entries $[h_0]_\alpha \overset{\text{iid}}{\sim} \mathcal{N}(0,1)$,*

- *the weight matrices are distributed as $A^i \overset{\text{iid}}{\sim} \frac{1}{\sqrt{Np_N}} W \odot B$ with $W$ and $B$ independent matrices having entries $[W]_{\alpha,\beta} \overset{\text{iid}}{\sim} \mathcal{N}(0,1)$ and $[B]_{\alpha,\beta} \overset{\text{iid}}{\sim} Ber(p_N)$,*

- *the sparsity parameter $p_N$ satisfies $Np_N \to \infty$ as $N \to \infty$,*

*achieves maximal probabilistic expressivity for compact sets.*

*Proof.* Following Cirone et al. [22, Section B.3.5], we only need to prove a bound of type

$$\left\| \frac{1}{N} \langle A_I h_0, A_J h_0 \rangle_{\mathbb{R}^N} - \delta_{I,J} \right\|_{L^2(\mathbb{P}_N)} \leq (\kappa(|I| + |J|))!! \; \mathcal{O}\left(\frac{1}{\sqrt{N}}\right) \tag{53}$$

as in the dense Gaussian case. That such sparse matrices present the same bounds as dense Gaussian ones follows from [21, Section 6.2], where it is shown that the bounds can only differ by a correction term of order $\mathcal{O}_{I,J}(\frac{1}{\sqrt{N}})$ where the constants are bounded by the number of pairings of $I \cup J \cup I \cup J$. $\square$

*Remark* B.3. Following Cirone et al. [21, Section 6.1], it is possible to prove that $W$ can be taken as having i.i.d. entries from a centred, symmetric but heavy tailed distribution given finiteness of even moments.

## B.3  Walsh–Hadamard Matrices

**Proposition B.4.** *The sequence of triplets $(\mathcal{A}_N, \mathbb{R}^N, \mathbb{P}_N)$ where $\mathcal{A}_N$ and $\mathbb{P}_N$ are such that*

- $\mathcal{A}_N := \{W \, diag(\Delta) : \Delta \in \mathbb{R}^N, \; W \in \mathbb{R}^{N \times N}, \; WW^\top = I_N\}$,

- *the initial value has independent standard Gaussian entries $[h_0]_\alpha \overset{\text{iid}}{\sim} \mathcal{N}(0,1)$,*

- *the weight matrices are distributed as $A^i \overset{\text{iid}}{\sim} \frac{1}{\sqrt{N}} H \, diag(\Delta)$ for a fixed $H \in \mathbb{R}^{N \times N}$ satisfying $HH^\top = NI_N$ and having entries bounded uniformly in $N$ by a constant $C$, and $\Delta \in \mathbb{R}^N$ having entries $[\Delta]_\alpha \overset{\text{iid}}{\sim} \mathcal{N}(0,1)$,*

*achieves maximal probabilistic expressivity for compact sets. In particular one can choose $H$ to be a Walsh–Hadamard matrix of order $N$ for computational efficiency.*

*Proof.* Following Cirone et al. [22, Section B.3.5], we only need to prove a bound of type

$$\left\| \frac{1}{N} \langle A_I h_0, A_J h_0 \rangle_{\mathbb{R}^N} - \delta_{I,J} \right\|_{L^2(\mathbb{P}_N)} \leq (\kappa(|I| + |J|))!! \; \mathcal{O}\left(\frac{1}{\sqrt{N}}\right). \tag{54}$$

We will place ourselves in the graphical setting of [21] and leverage the fact that ([21, Section 7.1]) their results and techniques hold even when the vertices are fixed to random vectors.

The first step is to notice that for $x \in \mathbb{R}^N$ one has the equivalence $W \, diag(\Delta) \cdot x = W \cdot (\Delta \odot x)$ which can be represented graphically as in Figure 2.

Figure 2: **Graphical representations of the matrix $W \, diag(\Delta)$.** Here $\ddagger$ edges correspond to identity matrices.

This leads to the product graph representation $G_{I,J}$ for $\frac{1}{N} \langle A_I h_0, A_J h_0 \rangle_{\mathbb{R}^N}$, where $I = i_1 \dots i_n$ and $J = j_1 \dots j_m$, given in Figure 3, which allows us to use [21, Proposition 1] to obtain the bounds.

$$\frac{1}{N}\langle A_I h_0,\, A_J h_0\rangle \;\equiv\; \frac{1}{N}\frac{1}{N^{\frac{n+m}{2}}} \quad$$ 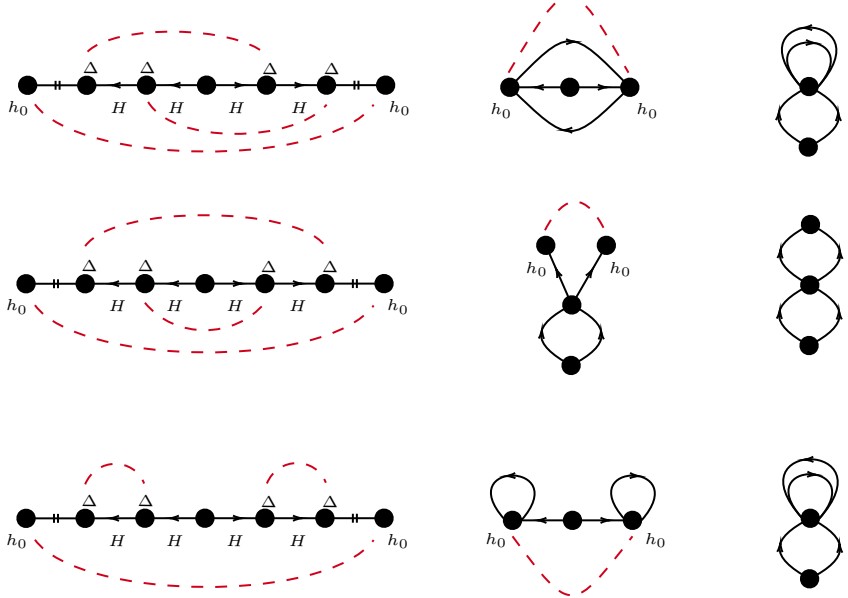

Figure 3: **The product $\frac{1}{N}\langle A_I h_0, A_J h_0\rangle_{\mathbb{R}^N}$ as a product graph G.**

In the present setting, the vertices labelled $h_0$ must be identified, and the remaining decorated vertices must be paired such that $\Delta_a$ is identified with $\Delta_b$ only if $a = b$. Each of these admissible pairings $\phi$ produces a graph $(G_{I,J})_\phi$ in which each vertex is assigned the vector of ones. Note that since $|[H]_{\alpha,\beta}| \le C$, one has $|H^{\odot k}| \le C^k$. The procedure is shown in Figure 4 for all pairings when $I = J = 11$.

Figure 4: **Construction of $(G_{I,J})_\phi$.** Here, we display all the pairings, represented by the red dashed lines, for $I = J = 11$ along with their intermediate stages. For simplicity, we omit the $H$ labels from edges with arrows.

Under these boundedness assumptions, note how leading-order pairings of $G_{I,J}$ are the ones having the maximum number of surviving vertices, meaning having $\frac{|I|+|J|}{2} + 1$ vertices. This can happen iff $I = J$, and even in this case, there exists only one pairing (the middle one in Figure 4) for which it holds that $(G_{I,I})_\phi \equiv \frac{1}{N}\frac{1}{N^{\frac{2|I|}{2}}} N^{|I|+1} = 1$. To see this, note that only the "middle" vertex of $G_{I,J}$ is not paired, so to get $\frac{|I|+|J|}{2} + 1$, all other vertices have to be identified in couples. This implies that the left and right adjacent vertices (call them $v_l$ and $v_r$) must be paired together: in fact the sub-graph comprising the middle vertex and the two adjacent edges corresponds to the matrix $HH^\top = N I_N$, hence we can remove this sub-graph, identify the adjacent vertices and take the factor $N$ in front without changing the value of the graph; but then if $v_l$ and $v_r$ were paired with other vertices we would identify at least 4 vertices. Hence $i_1$ has to be equal to $j_1$ and $v_l$ and $v_r$ must be identified. Proceeding by induction, from the middle out, we see that only the identity pairing has non-vanishing value.

To conclude we are left to prove that any pairing $\psi$ of $G_{I,J} \sqcup G_{I,J}$ not inducing the identity one on any of the two copies produces $|(G_{I,J} \sqcup G_{I,J})_\psi| \le \frac{1}{N}$, since the number of these pairings is less than the total number of pairings of the $\Delta$-labelled vertices of $G_{I,J} \sqcup G_{I,J}$, which is $(2(|I| + |J|))!!$. Then the inequality would hold with $\kappa = 2C^4$, since from the definition of product-graph, we see that the bound

$$|(G_{I,J} \sqcup G_{I,J})_\psi| \le \frac{1}{N^2}\frac{1}{N^{2\frac{|I|+|J|}{2}}} N^{V_\psi} C^{2(|I|+|J|)} \tag{55}$$

must hold, where $V_\psi$ is the number of vertices in $(G_{I,J} \sqcup G_{I,J})_\psi$. Thus it suffices to show $V_\psi \leq |I| + |J| + 1$.

To see this, notice that in any case $V_\psi \leq |I| + |J| + 2$ and that $V_\psi = |I| + |J| + 2$ iff $\psi$ identifies the random vertices in pairs. Once again there are special vertices which are not paired, the "middle" ones, the sub-graphs containing them correspond to the matrix $HH^\top = NI_N$ and can thus be removed by identifying the adjacent vertices and taking the $N$ factor out, so these vertices must be paired between themselves, and so on. This shows that $\psi$ has to separately pair both copies of $G_{I,J}$ in $G_{I,J} \sqcup G_{I,J}$ with identity pairings, but then such a $\psi$ would not be an atom-free pairing! Hence such a $\psi$ cannot exist and $V_\psi \leq |I| + |J| + 1$ always holds. $\qquad\square$

*Remark* B.5. Just as in the sparse case, and following [21, Section 6.1], it is possible to prove that the $\Delta$ matrices can be taken to have i.i.d. entries drawn from a centered, symmetric, but possibly heavy-tailed distribution, provided that the even moments are finite. This distributional adjustment is a useful technique for controlling the eigenvalue distribution of $\frac{1}{\sqrt{N}}H\mathrm{diag}(\Delta)$, ensuring it has favourable computational properties while providing a theoretical guarantee of preserving expressive power.

## B.4 Block Diagonal

**Proposition B.6.** *The sequence of triplets* $(\mathcal{A}_N, \mathbb{R}^N, \mathbb{P}_N)$ *where* $\mathcal{A}_N$ *and* $\mathbb{P}_N$ *are such that*

- $\mathcal{A}_N := \{\mathrm{BlockDiag}(B_1, B_2, \ldots, B_{k_N}) : b_N = \lceil \log(N) \rceil, k_N = \lceil N/b_N \rceil, B_h \in \mathbb{R}^{b_N \times b_N}\}$,

- *the initial value has independent standard Gaussian entries* $[h_0]_\alpha \overset{\text{iid}}{\sim} \mathcal{N}(0, 1)$,

- *the weight matrices are distributed as* $[B_i]_{\alpha,\beta} \overset{\text{iid}}{\sim} \frac{1}{\sqrt{b_N}}\mathcal{N}(0, 1)$,

*achieves maximal probabilistic expressivity for compact sets.*

*Proof.* Following Cirone et al. [22, Section B.3.5], we only need to prove a bound of type

$$\left\| \frac{1}{N} \langle A_I h_0, A_J h_0 \rangle_{\mathbb{R}^N} - \delta_{I,J} \right\|_{L^2(\mathbb{P}_N)} \leq (\kappa(|I| + |J|))!! \, \mathcal{O}\left(\frac{1}{\sqrt{N}}\right). \tag{56}$$

It suffices to notice that

$$\frac{1}{N} \langle A_I h_0, A_J h_0 \rangle_{\mathbb{R}^N} = \frac{1}{k_N} \sum_{l=1}^{k_N} \frac{1}{b_N} \langle B_l^I h_{0;l}, B_l^J h_{0;l} \rangle_{\mathbb{R}^{b_N}} ,$$

where $h_{0;l} := [h_0]_{(l-1)b_N+1,\ldots,lb_N}$, since we then know that

$$\left\| \frac{1}{b_N} \langle B_l^I h_{0;l}, B_l^J h_{0;l} \rangle_{\mathbb{R}^{b_N}} - \delta_{I,J} \right\|_{L^2(\mathbb{P}_N)} \leq (\kappa(|I| + |J|))!! \, \mathcal{O}\left(\frac{1}{\sqrt{b_N}}\right). \tag{57}$$

From the sum and the initial factor $\frac{1}{k_N}$ we obtain

$$\left\| \frac{1}{N} \langle A_I h_0, A_J h_0 \rangle_{\mathbb{R}^N} - \delta_{I,J} \right\|_{L^2(\mathbb{P}_N)} \leq (\kappa(|I| + |J|))!! \, \mathcal{O}\left(\frac{1}{\sqrt{b_N}}\right). \tag{58}$$

$\qquad\square$

## C  Log Linear Neural Controlled Differential Equations

Combining NCDEs with the Log-ODE method has been shown to produce state-of-the-art performance on a range of multivariate time series modelling benchmarks [12, 70, 98]. The same approach can be applied to Linear NCDEs (LNCDEs), and we refer to this approach as Log-LNCDEs. For a full introduction to the Log-NCDE method see [11, Section 3.2.2] and [98]. Here, we briefly outline

the application of the Log-ODE method to an LNCDE, assuming familiarity with the tensor product $\otimes$ and tensor algebra.

Recall the LNCDE model (4),

$$h_t = h_{t_0} + \int_{t_0}^t \sum_{i=1}^{d_\omega} A_\theta^i h_s \mathrm{d}\omega_s^i, \tag{59}$$

where $A_\theta^i$ is the linear vector field for each channel and $\omega_s^i$ are the channels of our control path. The log-signature of $\omega$ on $[s, t]$ is

$$\log(S(\omega)_{[s,t]}) = \log(1 + \mathbf{x}) = \sum_{n=1}^{\infty} \frac{(-1)^n}{n} \mathbf{x}^{\otimes n} \in \mathfrak{L}((\mathbb{R}_\omega^d)), \tag{60}$$

where $\mathbf{x} = (0, S^{\mathbf{1}}(\omega)_{[s,t]}, S^{\mathbf{2}}(\omega)_{[s,t]}, \ldots)$, $S^{\mathbf{j}}(\omega)_{[s,t]}$ is the collection of all signature components with words of length $j$, and $\mathfrak{L}((\mathbb{R}_\omega^d))$ is the free Lie algebra generated by $\mathbb{R}_\omega^d$ [14, 80]. From now, we consider the log-signature truncated at level $N$, which lives in the truncated free Lie algebra $\mathfrak{L}^N(\mathbb{R}_\omega^d)$. A basis for the truncated free Lie algebra is the Hall basis, which consists of up to the $(N-1)^{\text{th}}$ iterated Lie brackets of the basis of $\mathbb{R}_\omega^d$, denoted $\{e_k\}_{k=1}^{d_\omega}$, where the product is the tensor product [41]. Let $\{\hat{e}_k\}_{k=1}^{\beta(d_\omega, N)}$ denote the Hall basis of the truncated free Lie algebra, where $\beta(d_\omega, N)$ is the dimension of the truncated log-signature, and let $\lambda_k$ be the corresponding components of the log-signature.

Since $\mathfrak{L}^N(\mathbb{R}_\omega^d)$ is the truncated free Lie algebra, the linear map from the increments of the control to the linear ODE in (59) defined by

$$\sum_{i=1}^{d_\omega} A_\theta^i \mathrm{d}\omega_s^i \tag{61}$$

extends to a Lie algebra homomorphism acting on the log-signature of $\omega$ defined by

$$\sum_{i=1}^{\beta(d_\omega, N)} \bar{A}_\theta^k \lambda_k, \tag{62}$$

with

$$\bar{A}_\theta^k = A_\theta^k \tag{63}$$

for $1 \leq k \leq d_\omega$ and

$$\bar{A}_\theta^k = \bar{A}_\theta^j \bar{A}_\theta^i - \bar{A}_\theta^i \bar{A}_\theta^j \tag{64}$$

when the basis element $\hat{e}_k$ corresponds to the lie bracket of $\hat{e}_i$ and $\hat{e}_j$ [80]. The Lie bracket of $\bar{A}_\theta^i$ and $\bar{A}_\theta^j$ carries this sign as we are considering them as vector fields on $\mathbb{R}^d$. Similarly to a Log-NCDE, the Log-ODE method is applied to (59) over intervals $t_0 = r_0 < \ldots < r_m = t_n$ with $m < n$, such that

$$\mathrm{d}\tilde{h}_s = \left( \sum_{k=1}^{\beta(d_\omega, N)} \bar{A}_\theta^k \lambda_k^l \right) \tilde{h}_s, \tag{65}$$

for $s \in [r_l, r_{l+1}]$, where

$$\frac{\log(S^N(\omega)_{[r_l, r_{l+1}]})}{r_{l+1} - r_l} = \sum_{k=1}^{\beta(d_\omega, N)} \lambda_k^l \hat{e}_k. \tag{66}$$

The approximate solution is given by

$$\tilde{h}_{r_{l+1}} = \exp\left( \sum_{k=1}^{\beta(d_\omega, N)} \bar{A}_\theta^k \lambda_k^l \right) \tilde{h}_{r_l}, \tag{67}$$

and applying an associative scan has a reduced I/O cost as only $m < n$ state-transition matrices need to be materialised in GPU memory, as discussed in Section 4.6.

When applying the Log-ODE method to an LNCDE, the Lie brackets are calculated using the products of the linear vector fields for each channel, as opposed to the forward-mode auto-differentiated Jacobian-vector products of the vector field for Log-NCDEs. This significantly reduces the computational cost. For SLiCEs, the Log-ODE method is most beneficial when the linear ODE produced has the same vector field structure as the original vector fields. This is true for both diagonal and block-diagonal SLiCEs.

# D Additional experimental details and results

Single runs for all experiments can be completed on a 24GB NVIDIA RTX 4090 GPU in less than 24 hours. We use the following publicly available datasets, libraries, and baseline models:

- $A_5$ **Benchmark** [66]. License: MIT.
  URL: `https://github.com/jopetty/word-problem`
- **Formal Language Benchmark** [31]. License: CC-BY-4.0.
  URL: `https://arxiv.org/abs/2207.02098`
- **UEA Multivariate Time Series Classification Archive** [4]. License: GPL-3.0.
  URL: `https://www.timeseriesclassification.com/`, `https://github.com/time-series-machine-learning/tsml-repo`
- **S4D** [40]. License: Apache 2.0.
  URL: `https://github.com/state-spaces/s4`
- **Mamba** [39]. License: Apache 2.0.
  URL: `https://github.com/state-spaces/mamba`
- **DeltaNet, Gated DeltaNet, DeltaProduct** [85, 100, 101, 88]. License: MIT.
  URL: `https://github.com/fla-org/flash-linear-attention`
- **xLSTM / mLSTM / sLSTM** [6]. License: Apache 2.0.
  URL: `https://github.com/NX-AI/xlstm`
- **RWKV-7** [76]. License: Apache 2.0.
  URL: `https://github.com/BlinkDL/RWKV-LM`
- **Log-NCDE** [98]. License: CC-BY-4.0
  URL: `https://github.com/Benjamin-Walker/log-neural-cdes`
- **JAX** [10]. License: Apache 2.0.
  URL: `https://github.com/google/jax`
- **PyTorch** [75]. License: BSD Style, see here `https://github.com/pytorch/pytorch?tab=License-1-ov-file`
  URL: `https://github.com/pytorch/pytorch`

## D.1 The $A_5$ benchmark

The $A_5$ benchmark examines a model's state-tracking ability by asking the model to compose a sequence of even permutations on five elements [66]. There are 60 elements in the group, and the task is to compose between 3 and 20 elements. Our experiments follow the approach of Merrill et al. [66]. Models are trained using a token-tagging loss for 100,000 steps with a batch size of 256. For all sequence lengths, a small batch of sequences of length 2 are included at each training step to aid convergence. All models use Adam [54] with weight decay as their optimiser, and linear warm-up followed by cosine annealing with a minimum learning rate of $10^{-5}$ and a maximum learning rate of $10^{-3}$. Additionally, all models use dropout [90] at a rate of 0.1 and a trainable embedding layer.

The baseline models (Mamba, LSTM, mLSTM, sLSTM, and DeltaProduct) all use a hidden dimension of 1024. LSTM uses direct stacking whereas the other baseline models use stacked blocks consisting of a sequence model, a GLU layer [30], and layer normalisation [3]. DeltaProduct uses both gating and negative eigenvalues [101, 38]. All of the SLiCEs use

$$\frac{\omega_{t_{j+1}}^X - \omega_{t_j}^X}{t_{j+1} - t_j} = (1, X_{t_j}) \tag{68}$$

and 1024 non-zero parameters for each $A_\theta^i$. For the diagonal and Walsh–Hadamard SLiCEs, this corresponds to a hidden dimension of 1024, and for the dense SLiCE, this corresponds to a hidden dimension of 32. The DPLR SLiCE uses a rank of 2, giving a hidden dimension of 205, the block-diagonal SLiCE uses $b_i = 4$, giving a hidden dimension of 256, and the diagonal-dense SLiCE uses a dense block size of 23, giving a hidden dimension of 518. The sparse SLiCE uses a hidden dimension of 128 and a sparsity of $\epsilon = \frac{3}{7}$. All SLiCEs use stacked blocks consisting of a SLiCE, a linear layer followed by a tanh activation function, layer normalisation, and weight regularisation. Due to issues with convergence, Walsh–Hadamard's diagonal matrix $D_\theta^i$ is parametrised to take values between $-1$ and 1.

## D.2 Regular language tasks

Delétang et al. [31] introduced four regular language tasks as part of the formal language benchmark:

1. **Cycle navigation.** Infer the final position of a walk on a cycle starting at the origin. Actions are randomly sampled from "go forward one step", "stay in the same place", and "go backward one step". We use a cycle of length 5, therefore a random guesser will achieve an accuracy of 20%.

2. **Even pairs.** There are two states in the system and the goal is to determine if there is an equal ("even") number of transitions between the two states. As each sequence is either "even" or not, a random guesser will achieve an accuracy of 50%.

3. **Modular arithmetic (no brackets).** Performs modular arithmetic without any brackets, i.e. only handling addition and multiplication. We use mod 5 and hence a random guesser will achieve 20%.

4. **Parity.** There are two elements in the system. To determine the parity, the number of the second element is counted to determine if it is even or odd. This can be viewed as modular summation with mod 2. A random guesser achieves 50%.

This benchmark tests the ability of models to length generalise on state-tracking tasks, by training models on sequences from length 3 to 40 and evaluating models on sequences from length 40 to 256. Following Beck et al. [6], all baseline models use two stacked layers. In addition to the hidden dimension of 512 used by Beck et al. [6], we also train the baseline models with a hidden dimension of 128, selecting the value that yields the highest average validation accuracy for each model on each task. For Mamba, which does not support a hidden dimension of 128, we instead choose between 256 and 512 based on validation performance. All models are trained using a token-tagging loss for 100,000 steps with a batch size of 256. All models use Adam [54] with weight decay as their optimiser, and linear warm-up followed by cosine annealing with a minimum learning rate of $10^{-5}$ and a maximum learning rate of $2 \times 10^{-3}$. Additionally, all models use dropout [90] with a rate of 0.01 and a trainable embedding layer.

Similarly to the $A_5$ benchmark, LSTM uses direct stacking whereas the other baseline models use stacked blocks consisting of a sequence model, a GLU layer [30], and layer normalisation [3]. The baseline models considered are vanilla DeltaNet [85, 99], DeltaNet with negative eigenvalues (DeltaNet[-1,1]) [38], Gated DeltaNet [101], Gated DeltaProduct with negative eigenvalues and a rank of 2, sLSTM [6], mLSTM [6], xLSTM [6], RWKV-7 [76], a Transformer [96], S4D[40], and Mamba [39].

We consider all SLiCEs on this benchmark except sparse due to the lack of an efficient implementation. All SLiCEs use

$$\frac{\omega_{t_{j+1}}^X - \omega_{t_j}^X}{t_{j+1} - t_j} = (1, X_{t_j}), \tag{69}$$

and two stacked blocks consisting of the sequence layer, a linear layer followed by a tanh activation function, and layer normalisation. For the diagonal and Walsh–Hadamard SLiCE, we consider hidden dimensions of 128 and 512, corresponding to 128 and 512 non-zero parameters per state-transition matrix, respectively. For all other SLiCEs, the number of non-zero parameters in the state-transition matrix is fixed at 512. For DPLR we consider ranks of $r = 1, 2, 4, 8$, and for block-diagonal we consider two variants, $b_i = b$ for all $i$ with $b = 2, 4, 8, 16$, and $b_i = 1$ for $i = 1, \ldots, k-1$, and then a final dense block $b_k = b$ for $b = 2, 4, 8, 16$, referred to as diagonal-dense SLiCE (D-DE-SLiCE). Due to issues with convergence, Walsh–Hadamard's diagonal matrix $D_\theta^i$ is parametrised to take values between $-1$ and 1.

Table 4 reports the average validation accuracy for all SLiCEs. Although replacing the diagonal SLiCE with a Walsh–Hadamard SLiCE improves performance on the $A_5$ benchmark, it decreases the average validation accuracy on regular language tasks. We hypothesise that this degradation arises from the same factors responsible for the Walsh–Hadamard SLiCE's poor performance on the $A_5$ length generalisation task. For a block-diagonal SLiCE with a fixed block size, any $b_i > 1$ leads to higher average validation accuracy than the diagonal SLiCE, with performance peaking at $b_i = 4$. This provides empirical evidence that, under a fixed computational budget (or a fixed number of non-zero state-transition parameters), there exists a trade-off between expressivity and hidden dimension. DPLR-SLiCE exhibits a similar pattern, achieving its highest accuracy at $r = 4$. Using a

Table 4: **Results of SLiCEs on formal language tasks.** Average and standard deviation of validation accuracy over five runs on the regular language tasks for SLiCEs with diagonal, Walsh–Hadamard (WH), block-diagonal (BD), diagonal-dense (D-DE), and DPLR structures.

| Model | Cycle Nav. | Even Pairs | Mod Arith. No Brack. | Parity | Average |
|---|---|---|---|---|---|
| Diagonal$_{d_h=128}$ | $69.5 \pm 6.3$ | $100.0 \pm 0.0$ | $20.8 \pm 0.2$ | $89.12 \pm 18.6$ | 69.9 |
| Diagonal$_{d_h=512}$ | $59.7 \pm 5.1$ | $100.0 \pm 0.0$ | $20.9 \pm 0.1$ | $100.0 \pm 0.0$ | 70.2 |
| WH$_{d_h=128}$ | $69.7 \pm 8.8$ | $93.1 \pm 13.9$ | $20.5 \pm 0.3$ | $50.7 \pm 0.3$ | 58.5 |
| WH$_{d_h=512}$ | $35.5 \pm 1.8$ | $58.5 \pm 2.5$ | $23.8 \pm 1.1$ | $71.4 \pm 12.9$ | 47.3 |
| BD$_{d_h=256,\,b=2}$ | $92.5 \pm 14.0$ | $72.7 \pm 3.0$ | $37.7 \pm 2.6$ | $99.6 \pm 0.8$ | 75.6 |
| BD$_{d_h=128,\,b=4}$ | $99.8 \pm 0.2$ | $85.9 \pm 11.3$ | $54.0 \pm 12.5$ | $95.3 \pm 3.9$ | 83.8 |
| BD$_{d_h=64,\,b=8}$ | $99.9 \pm 0.1$ | $91.3 \pm 6.3$ | $70.6 \pm 21.4$ | $54.1 \pm 4.9$ | 79.0 |
| BD$_{d_h=32,\,b=16}$ | $97.6 \pm 3.5$ | $94.7 \pm 6.5$ | $76.6 \pm 22.1$ | $50.8 \pm 0.3$ | 79.9 |
| D–DE$_{d_h=510,\,b=2}$ | $61.9 \pm 20.4$ | $91.3 \pm 11.5$ | $20.8 \pm 0.2$ | $97.8 \pm 2.9$ | 67.9 |
| D–DE$_{d_h=500,\,b=4}$ | $81.6 \pm 15.0$ | $85.3 \pm 18.0$ | $29.4 \pm 15.4$ | $83.7 \pm 9.4$ | 70.0 |
| D–DE$_{d_h=456,\,b=8}$ | $90.6 \pm 9.4$ | $90.7 \pm 3.0$ | $31.0 \pm 4.2$ | $79.9 \pm 3.5$ | 73.1 |
| D–DE$_{d_h=272\,b=16}$ | $73.3 \pm 29.4$ | $84.8 \pm 8.5$ | $98.4 \pm 0.7$ | $83.8 \pm 11.3$ | 85.1 |
| DPLR$_{d_h=171,r=1}$ | $46.5 \pm 26.3$ | $91.1 \pm 4.4$ | $25.8 \pm 10.2$ | $87.8 \pm 9.5$ | 62.8 |
| DPLR$_{d_h=102,r=2}$ | $53.1 \pm 14.7$ | $96.8 \pm 5.1$ | $43.9 \pm 9.0$ | $79.7 \pm 14.4$ | 68.4 |
| DPLR$_{d_h=57,r=4}$ | $81.1 \pm 16.6$ | $100.0 \pm 0.0$ | $68.3 \pm 19.3$ | $91.0 \pm 18.0$ | 85.1 |
| DPLR$_{d_h=30,r=8}$ | $90.1 \pm 10.2$ | $100.0 \pm 0.0$ | $60.3 \pm 19.7$ | $50.7 \pm 0.3$ | 75.3 |
| **Random** | 20.0 | 50.0 | 20.0 | 50.0 | 35.0 |

diagonal-dense SLiCE enables larger hidden dimensions while maintaining dense block connections. The largest dense block configuration, $b = 16$, attains the joint highest average accuracy along with the $r = 4$ DPLR-SLiCE. The strong performance of DPLR-SLiCE is consistent with that of Gated DeltaProduct model with negative eigenvalues, which serves as the best-performing baseline.

### D.3 UEA multivariate time series classification archive

The experiments on the UEA multivariate time series classification archive follow the approach of Walker et al. [98], using the same data splits. To mitigate convergence issues, the time series were scaled to the range $[-1, 1]$. The log-signatures were scaled down by a factor of ten on Heartbeat for all structures except DPLR, and by a factor of one hundred on Heartbeat and MotorImagery for DPLR. All SLiCE models use the same hyperparameters as the Log-NCDE, differing only in that the non-linear vector field is replaced by their respective structured linear vector fields. The block-diagonal structure uses $b_i = 4$, the diagonal-dense structure uses a dense block of dimension 16, the DPLR structure uses a rank of 4, and the sparse structure uses a sparsity of $0.1$. The Log-ODE method is applied over the same intervals as in the Log-NCDE. The flows are composed using an associative parallel scan applied to chunks of size 128, with each chunk processed recurrently. All SLiCEs take $\omega_s^X = X_s$.

Table 5 presents a breakdown of the average test accuracies from Table 3 across individual datasets. Figure 5 provides a visual representation of the performance of the baseline models, dense LNCDE, diagonal SLiCE, and block-diagonal SLiCE on the UEA-MTSCA benchmark. Although Log-NCDEs improve the average time per training step compared with NCDEs and NRDEs, a substantial gap remains relative to S5, S6, LRU, and Mamba. Replacing the non-linear vector field with a block-diagonal linear vector field reduces the average time per training step by a factor of 20, bringing it within the same order of magnitude as the SSM baselines without degrading the average test accuracy. Using a diagonal linear vector field further decreases the training time to below that of the SSM baselines while maintaining comparable accuracy to the baselines. However, the average test accuracy is lower than that of the block-diagonal variant, consistent with its reduced expressivity. Employing a dense linear vector field slightly increases run-time and substantially raises GPU memory usage compared with the block-diagonal case. It also reduces the average test accuracy, which may indicate over-fitting due to the large number of parameters per state transition. This result suggests that using a block-diagonal linear vector field may have benefits beyond simply reducing the computational cost of the model.

Table 5: **UEA test accuracy (%) for all models across six datasets.** The best-performing model in each column is highlighted in bold, and the second-best is underlined. Lower is better for Average Rank.

| Model | EW | EC | HB | MI | SCP1 | SCP2 | Av. Acc | Av. Rank |
|---|---|---|---|---|---|---|---|---|
| BD-SLiCE | $86 \pm 4$ | $29 \pm 7$ | $77 \pm 6$ | $53 \pm 3$ | $85 \pm 2$ | $\mathbf{54 \pm 8}$ | $\underline{64.0}$ | $\mathbf{3.2}$ |
| Log-NCDE | $86 \pm 6$ | $\mathbf{34 \pm 7}$ | $75 \pm 5$ | $54 \pm 6$ | $\underline{83 \pm 3}$ | $\underline{54 \pm 5}$ | $64.3$ | $\underline{4.0}$ |
| D-DE-SLiCE | $86 \pm 6$ | $27 \pm 7$ | $74 \pm 4$ | $\mathbf{55 \pm 4}$ | $85 \pm 4$ | $\overline{52 \pm 5}$ | $63.0$ | $5.7$ |
| WH-SLiCE | $85 \pm 6$ | $30 \pm 5$ | $76 \pm 6$ | $49 \pm 7$ | $82 \pm 3$ | $52 \pm 6$ | $62.5$ | $6.7$ |
| DPLR-SLiCE | $84 \pm 6$ | $28 \pm 5$ | $74 \pm 6$ | $52 \pm 6$ | $84 \pm 3$ | $51 \pm 6$ | $62.0$ | $7.0$ |
| D-SLiCE | $79 \pm 6$ | $27 \pm 5$ | $73 \pm 6$ | $\underline{54 \pm 7}$ | $84 \pm 3$ | $53 \pm 6$ | $61.7$ | $7.2$ |
| LRU | $\underline{88 \pm 3}$ | $22 \pm 3$ | $\mathbf{78 \pm 7}$ | $\overline{48 \pm 6}$ | $83 \pm 4$ | $51 \pm 4$ | $61.7$ | $7.3$ |
| S6 | $85 \pm 17$ | $26 \pm 7$ | $77 \pm 9$ | $51 \pm 5$ | $83 \pm 3$ | $50 \pm 10$ | $62.0$ | $7.7$ |
| S5 | $81 \pm 4$ | $24 \pm 5$ | $\underline{78 \pm 6}$ | $48 \pm 6$ | $\mathbf{90 \pm 5}$ | $51 \pm 3$ | $61.8$ | $8.0$ |
| S-SLiCE | $\underline{88 \pm 5}$ | $30 \pm 7$ | $\overline{73 \pm 6}$ | $48 \pm 3$ | $83 \pm 2$ | $49 \pm 4$ | $61.8$ | $8.2$ |
| DE-LNCDE | $\mathbf{88 \pm 4}$ | $\overline{26 \pm 5}$ | $74 \pm 5$ | $50 \pm 6$ | $82 \pm 4$ | $50 \pm 3$ | $61.6$ | $8.3$ |
| NCDE | $75 \pm 4$ | $30 \pm 7$ | $74 \pm 3$ | $50 \pm 3$ | $80 \pm 6$ | $53 \pm 3$ | $60.2$ | $8.8$ |
| NRDE | $84 \pm 8$ | $25 \pm 2$ | $73 \pm 5$ | $47 \pm 6$ | $81 \pm 3$ | $\underline{54 \pm 7}$ | $60.6$ | $10.3$ |
| Mamba | $71 \pm 16$ | $28 \pm 5$ | $76 \pm 4$ | $48 \pm 5$ | $81 \pm 2$ | $\overline{48 \pm 4}$ | $58.6$ | $10.8$ |

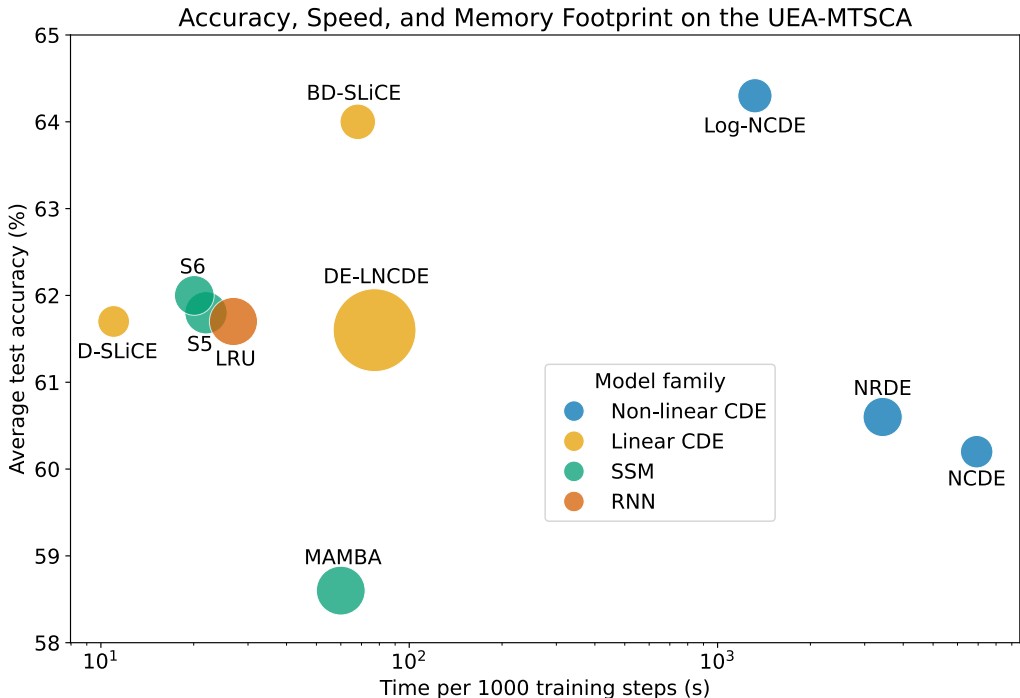

Figure 5: **Average per-step training time versus average validation accuracy across six multivariate time-series classification datasets from the UEA-MTSCA.** Each point represents a model, with circle area proportional to average GPU memory usage. We compare four families of models: a recurrent neural network (LRU), SSMs (S5, S6, and Mamba), non-linear NCDEs (NCDE, NRDE, and Log-NCDE), and linear NCDEs (Diagonal SLiCE, Block-Diagonal SLiCE, and Dense LNCDE). All test accuracy results except linear NCDEs are from Walker et al. [98]. All timing and GPU memory results were re-performed on an NVIDIA H100 GPU.

Table 6: **Training time for EigenWorms using a parallel associative scan without applying Log-ODE method.** Time per 1,000 training steps (s) for diagonal SLiCE, block-diagonal SLiCE, and dense LNCDE on EigenWorms when a parallel associative scan is applied with various chunk sizes. Experiments were performed on an NVIDIA H100, and the batch size is 1.

| Parallel Steps | D-SLiCE | BD-SLiCE | DE-LNCDE |
|---|---|---|---|
| None | 311.0 | 374.89 | 444.68 |
| 4 | 134.2 | 326.75 | 439.51 |
| 16 | 57.50 | 161.74 | 257.56 |
| 64 | 31.01 | 68.59 | 126.01 |
| 256 | 21.10 | 30.53 | 71.54 |

Table 7: **Comparison of training time and GPU memory for EigenWorms using a parallel associative scan and applying the Log-ODE method.** Experiments were performed on an NVIDIA H100 with a batch size of 4 for diagonal SLiCE, block-diagonal SLiCE, and dense LNCDE.

| Model | Metric | Log-ODE Interval | Parallel Steps | |
|---|---|---|---|---|
| | | | None | 128 |
| D-SLiCE | Time / 1k steps [s] | 1 | 317.49 | 23.51 |
| | | 12 | 33.03 | 10.96 |
| | GPU Memory [GB] | 1 | 2.69 | 2.69 |
| | | 12 | 2.69 | 2.69 |
| BD-SLiCE | Time / 1k steps [s] | 1 | 378.20 | 48.74 |
| | | 12 | 46.98 | 13.24 |
| | GPU Memory [GB] | 1 | 2.69 | 4.73 |
| | | 12 | 2.69 | 2.69 |
| DE-LNCDE | Time / 1k steps [s] | 1 | 465.94 | 167.86 |
| | | 12 | 47.95 | 29.37 |
| | GPU Memory [GB] | 1 | 35.45 | 51.84 |
| | | 12 | 2.69 | 6.79 |

To conclude, we evaluate how the Log-ODE method and parallel associative scan influence time per training step and GPU memory for the diagonal SLiCE, block-diagonal SLiCE, and dense LNCDE. The EigenWorms dataset is chosen for this comparison, as it contains approximately 18,000 observations per time series. Table 6 summarises the effect of applying a parallel associative scan with varying chunk sizes on time per 1,000 training steps without the Log-ODE method. For all three models, increasing the number of parallel steps yields strong reductions in time per training step. The impact of the high I/O costs associated with an associative scan is evident from the diminishing benefit of a small number of parallel steps as you move from diagonal, through block-diagonal, to dense matrices.

Table 7 compares the time per 1,000 training steps and GPU memory for diagonal SLiCE, block-diagonal SLiCE, and dense LNCDE on EigenWorms when using a parallel associative scan and the Log-ODE method. As expected, both GPU memory and run-time increase monotonically for every combination of associative scan and Log-ODE method as the model structure transitions from diagonal, through block-diagonal, to dense matrices. The consistent 2.69 GB floor across several configurations likely reflects peak memory usage from fixed operations outside the recurrence. Without the Log-ODE method, the dense LNCDE exhibits high GPU memory consumption, which constrained experiments to a batch size of 4. When both the Log-ODE method and a parallel associative scan are applied, the diagonal and block-diagonal SLiCE achieve comparable times per training step, indicating that the recurrence contributes less to overall computation under these parameter settings. Overall, combining the Log-ODE method with a parallel associative scan reduces the time per training step by $\sim 30\times$ for the diagonal and block-diagonal SLiCE without affecting GPU memory, and by $\sim 16\times$ for the dense LNCDE while lowering GPU memory usage by over $5\times$.

