# OpenReview forum: "Structured Linear CDEs: Maximally Expressive and Parallel-in-Time Sequence Models"
_NeurIPS.cc/2025/Conference — NeurIPS 2025 spotlight_

### Official Review · Reviewer_EATe · 2025-06-26

**Clarity:** 3
**Significance:** 2
**Originality:** 3
**Rating:** 5
**Confidence:** 4

**Summary:**

The paper introduces Structured Linear CDEs (SLiCEs), a unified framework that captures and extends various structured state-transition matrices (block-diagonal, sparse, Walsh-Hadamard, etc.) under linear Linear Neural Controlled Differential Equations (Linear NCDEs or LNCDEs), providing asymptotic expressivity guarantees in the infinite-width limit. The authors prove that diagonal LNCDEs are not universal, whereas sparse, block-diagonal and Walsh-Hadamard structures are. Empirically, SLiCEs match dense NCDEs on synthetic benchmarks, confirming the predictions by their theory.

**Questions:**

- Does the first-order approximation of the exponential in Eq. 13 not compromise numerical stability or accuracy on long intervals compared to higher-order Log-ODEs? The text just refers to "common practice" but no further explanation or justification is given for this approximation.
- Are the S-LNCDEs the only model variant that doesn't have an efficient implementation (and would therefore benefit from dedicated hardware), or do you see potential for further improvement in other variants, too?
- There is no evaluation of eigenvalue distributions or stability of structured $A_i$ over long horizons, though state-tracking depends on avoiding blow-up or vanishing modes. How is this handled by your theory and in your experiments?

**Ethical Concerns:**

["NO or VERY MINOR ethics concerns only"]

**Final Justification:**

The additional experimental results are convincing and all my comments and questions have been fully addressed

**Limitations:**

yes

**Quality:**

3

**Strengths And Weaknesses:**

**Strengths**:
- Introduces SLiCEs as a unified framework to subsume and extend a wide range of structured, input-dependent state-transition matrices (block-diagonal, sparse, Walsh–Hadamard, DPLR, diagonal) under a common LNCDE lens, clarifying connections across prior work and novel variants.
- Proves maximal probabilistic expressivity for block-diagonal (Thm 4.1), sparse (Thm 4.2), and Walsh–Hadamard (Thm 4.3) structures, matching dense matrices in the limit. This is an interesting and insightful finding for the community to build upon.
- Evaluates on (1) $A_5$ state-tracking (which clearly delineates expressive and non-expressive SLICEs as predicted by the theory), (2) formal-language length's generalisation, and (3) various multivariate tasks (showing block-diagonal SLiCE matches Log-NCDE accuracy while cutting per-step time by ~20x).
- Offers a promising route for block-diagonal SLICEs.
- While the theory and experiments are limited to vector-valued states, a derivation for matrix-valued states is provided.

**Weaknesses**:
- Theoretical results are only in the limit of infinite hidden dimension, which does not clearly translate to practical implementations with finite hidden dimensions. Besides a computational complexity analysis and results on toy tasks (which are convincing, but still synthetic), there is little guidance (or predictions) for practical applications of these models.
- Results for the sparse LNCDE variant are not very convincing, as they are only evaluated on the A5 benchmark and not on the other two benchmarks. Moreover, it was only evaluated for a fixed sparsity factor of 3/7 which was chosen arbitrarily (Appendix D.1, no further explanation). There is no discussion on how the sparsity factor influences the expressivity of the resulting S-LNCDE model. A similar analysis to that presented in Appendix D.2 would strengthen these results. It would further be interesting to connect these results to existing work on sparsity in LNCDEs such as [1] on sparse inputs with SSMs and [2] on unstructured sparsity in S5, which relates closely to the proposed S-LNCDE and presents hardware that appears to accelerate unstructured sparsity.
- For the BD-LNCDE, block size $b$ and rank $r$ critically affect performance (e.g. large $b$ yields poor `Parity` accuracy), yet the analysis is confined to appendix, and no clear practical guidelines are given.
- In the matrix-valued state derivation, since each column $h^k_t$ follows its own copy of the same LNCDE (plus bias), there is no mechanism for $H$’s columns to interact—mixing between columns is prohibited unless one uses a full $d_hn\times d_hn$ coupling, which would blow up parameters.

Minor
- relation of this work to sparsity through pruning and the lottery ticket hypothesis is not quite clear from the related work section.
- Figure 1's caption refers to a "gray region" that is not gray, perhaps better to just call it the "shaded region"
- is Appendix B referenced in the main text?
- it would be helpful for the reader to also have the Log-LNCDE added to Table 1, especially for the memory-compute tradeoff that is mentioned in Section 5.3.

[1] [Schöne et al, 2024](https://arxiv.org/abs/2411.12603)

[2] [Pierro et al, 2025](https://openreview.net/forum?id=UNrfYfbLZ3)

---

> ### Author Rebuttal · Authors · 2025-07-31
>
> Thank you very much for your comprehensive review.
>
> ## Weaknesses
>
> 1. We have expanded our experiments on the UEA benchmark, which consists of real-world time series datasets, to include all of the SLiCE structures. Furthermore, we have performed empirical runtime and GPU memory experiments on EigenWorms, the longest dataset considered in this paper, to provide additional guidance when applying these models in practice. For more details, please see the section of our response to reviewer 7XXo titled "Questions 1, 3, and 4". We hope these additions partly address your concerns, although we acknowledge that there remains open work on understanding the behaviour of these models at finite hidden dimension and how to choose their hyperparameters.
> 2. On the A5 benchmark, all of the SLiCE architectures were chosen to have \$1024\$ non-zero parameters per state-transition matrix (Lines 241, 802). We were computationally limited to a hidden dimension of \$128\$ for dense matrices, and so chose a hidden dimension of \$128\$ and a sparsity factor of \$3/7\$ for S-LNCDE, giving \$1024\$ non-zero parameters. We have amended the paper to make this decision clearer. We have also added an additional experiment on the A5 benchmark to investigate the impact of sparsity. We trained the S-LNCDE on the length \$20\$ A5 dataset and varied the sparsity factor \$\epsilon\$. We find that as \$\epsilon\$ decreases, the number of training steps taken to converge increases, but the model still converges to $100\\%$ validation accuracy with only $1\\%$ of the parameters being non-zero. Thank you for bringing these two sparse papers to our attention, we will incorporate them into our discussion.
> 3. We haved moved part of the discussion on the impact of block size and rank to the main body of the text.
> 4. We note that many existing sequence-model architectures that use matrix-valued hidden states, including Gated Linear Attention, RetNet, RWKV6, mLSTM, and Mamba, do not have a method for column interaction in the recurrence. Furthermore, if desired, column mixing can be achieved by right-multiplying by an input-dependent \$n \times n\$ matrix, as is done in DeltaNet, without requiring a full \$d\_h n \times d\_h n\$ coupling. We have expanded Appendix C to broaden the comparison with existing matrix-valued hidden-state models.
>
> ## Minor
>
> 1. We have modified Lines 56–60 to the following (citations removed for brevity):
> “Utilising structured matrices to reduce the computational burden of neural networks extends beyond sequence models. The lottery-ticket hypothesis argues that dense networks contain sparse sub-networks that, when trained in isolation, can match the accuracy of the full model. Such sub-networks have been uncovered by pruning before, during, and after training. SLiCEs differ from pruning by imposing structured sparsity at initialisation and training the resulting sparse model directly. Other structured-matrix approaches include sparse Transformers, 2:4 sparsity in linear layers, and Monarch layers, which factorise weight matrices into two block-diagonal components.”
>
> 2. We have amended the caption to “shaded region”.
>
> 3. Yes, Appendix B is referenced on Line 218.
>
> 4. We have added the following line to **Table 1**:
>
>    | **Model**   | **Parameters**        | **Recurrent Cost**                                     | **Parallel Cost**                                              |
>    | ----------- | --------------------- | ------------------------------------------------------ | -------------------------------------------------------------- |
>    | Log-X-LNCDE | \$\mathcal{O}(P\_X)\$ | \$\mathcal{O}\left(\tfrac{R\_X}{s}d\_h^{N-1}\right)\$ | \$\mathcal{O}\left(\log\tfrac{n}{s},C\_X d\_h^{N-1}\right)\$ |
>
>    *Log-X-LNCDE corresponds to applying the Log-ODE method with intervals containing \$s\$ samples and a truncation depth of \$N\$, where X corresponds to a specific SLiCE structure (DE, D, DPLR, S, WH, or BD), \$P\_X\$ is the order of parameters for X, \$R\_X\$ is the order of the recurrent cost for X, and \$C\_X\$ is the order of the cost per composition for X.*
>
>    We have also expanded the discussion in Section 5.3 to contrast the theoretical computational cost of the Log-ODE method with the reduction in I/O costs. Empirically, we show that the Log-ODE method leads to practical speed-ups for the BD-LNCDE on the longest dataset we consider, the EigenWorms dataset, which has almost \$18{,}000\$ observations per sequence. See the table below for details.
>
>    | Metric                    | Log-ODE Interval | Recurrent | Parallel steps 128 |
>    | ------------------------- | ---------------- | ---------: | -----------: |
>    | **Time / 1 k steps [s]** | 1                |     227.26 |        52.58 |
>    |                           | 12               |      24.66 |        13.40 |
>    | **GPU Memory [GB]**      | 1                |       3.48 |         7.59 |
>    |                           | 12               |       3.48 |         3.49 |
>
> ## Questions
>
> 1. Yes, although we did not find that first-order approximation compromised training stability in practice. We have expanded the discussion to mention that this is done as a computational cost-saving measure, and we did not find training stability impacted by this choice.
> 2. The fast Walsh–Hadamard transform, although supported, would likely benefit from improved software and dedicated hardware. For example, the fastest inference method we are aware of (PyTorch’s Hadacore) does not support back-propagation.
> 3. Inspired by the \$\mathrm{Lip}(\gamma)\$ regularisation introduced in [1], all SLiCEs use weight regularisation on their state-transition matrices. The paper does not make this explicit and we apologise for this oversight. The main body and Appendix now discuss this. The Walsh–Hadamard LNCDE’s training was still sometimes unstable. Therefore, we constrained the range of the diagonal weights to $\[-1,1]\$, which means that the eigenvalues of the state-transition matrix have modulus \$\leq 1\$. This was briefly discussed in Appendix D, but we have now included an expanded discussion in Section 4.5.
>
> [1] Benjamin Walker, Andrew D. McLeod, Tiexin Qin, Yichuan Cheng, Haoliang Li, and Terry Lyons. *Log neural controlled differential equations: The lie brackets make a difference.* International Conference on Machine Learning, 2024.

---

> > ### Comment · Reviewer_EATe · 2025-08-05
> >
> > I thank the authors for their detailed response. The additional experimental results are convincing and all my comments and questions have been fully addressed. I will raise my score to 5.
> >
> > I would find it very helpful to include the results for varying levels of sparsity in the paper (or appendix), and to clarify what level of sparsity was chosen for the Table presented in the rebuttal to reviewer `7XXo` (I assume it is also $3/7$).

---

> ### Author Response · Authors · 2025-08-05
>
> Thank you very much for your thoughtful follow-up and for raising your score. We are glad our response addressed your concerns.
>
> We agree that including the results for varying levels of sparsity and clarifying that S-LNCDE used a sparsity of $\epsilon=3/7$ in the Table presented to reviewer 7XXo will strengthen the paper. We will incorporate these into the final version.

---

> > ### Comment · Reviewer_EATe · 2025-08-05
> >
> > Thank you for your quick response. All the best with your research!

---

### Official Review · Reviewer_oajU · 2025-06-28

**Clarity:** 2
**Significance:** 3
**Originality:** 3
**Rating:** 4
**Confidence:** 4

**Summary:**

This paper introduces the concept of linear neural control differential equations (LNCDE), which have input-dependent state transition matrices. The paper explores a theoretical framework of many variants of these with varying structures on their transition matrices, including diagonal, diagonal plus low rank (DPLR), block diagonal, and Hadamard, discussing their expressivity properties and efficiency characteristics. These methods are empirically validated on difficult synthetic tasks such as state tracking on the A5 permutation group.

**Questions:**

Several of my questions are listed above in the weaknesses section.

I think the paper would be more valuable to a broader ML audience if there were more intuitive comparisons to existing methods. For example, a direct contrast between the diagonal LNCDE with Mamba (showing specific equations and computation method, discussing all axes in which they differ, etc)

Additional minor comment: In several places, it is stated that S4 is a diagonal SSM (e.g. line 148). This is not entirely accurate as the original S4 is DPLR, but the more popularly used S4D is indeed diagonal. They should perhaps be clarified where appropriate.

**Ethical Concerns:**

["NO or VERY MINOR ethics concerns only"]

**Limitations:**

The work does not discuss limitations too much. It would be useful to contrast the proposed LNCDE methods more carefully to popular models in literature.

**Quality:**

3

**Strengths And Weaknesses:**

Strengths
- The paper introduces a useful theoretical framework on the expressivity of various sequence models, which is a very timely topic.
- The paper is technically sound, although it could benefit from unpacking some background and notation more.
- It introduces some novel families of methods, such as the Walsh-Hadamard variant.
- The methods have concrete empirical benefits, such as being able to solve difficult state tracking tasks, verifying their increased expressivity over popular recurrent models in the literature.

Weaknesses
- A number of implementation details are missing. For example, I found it hard to understand how the proposed L and C/D/E layers are incorporated into a neural architecture block outside of the core recurrence; most of the baseline architectures (e.g. Mamba, DeltaNet, Transformer etc.) have slight variations in their structure such as the dimensions, whether there is a multiplicative gate, the head dimensions and sharing structure, and so on.
- Similarly, I feel that details were missing from the implementation of the core layer. Figure 5 is surprising to me that Diagonal LNCDE is much faster than Mamba and block diagonal/dense are comparable to Mamba. Shouldn't diagonal be the same speed as Mamba since they are the same structure? What is the difference between Mamba and S6 here? Where is the Walsh-Hadamard method, is that the same speed as the Dense? I think the exact nomenclature differences between all these methods and how they relate to the surrounding neural network block structure and the exact dimensions and so on are not clear throughout the manuscript.
- Empirically, the experiments test sequential tasks that are not the most popular or impactful ones that the related work (such as Mamba, DeltaNet, xLSTM, etc.) care about. In particular, most of those methods were designed for language modeling, which was not evaluated in this paper.

---

> ### Author Rebuttal · Authors · 2025-07-31
>
> Thank you very much for your thorough review.
>
> ## Clarity
>
> We have redrafted the paper to improve the clarity. Changes to the abstract and introduction are detailed in our response to reviewer 7XXo, and details about a new introductory chapter in the Appendix are given in our response to reviewer g3ZB. Here, we focus on the changes we have made to improve the clarity of our model architecture and how it compares to existing models in the literature. We have added the following paragraph as Section 4.7 in the paper.
>
> ### Comparison to Existing Architectures
>
> SLiCEs are closely related to S6, the recurrent component of Mamba. In particular, if you remove the bias term (i.e. $B = 0$), then the continuous form of S6 is given by a LNCDE,
>
> $$
> dh_t = Dh_t \mathrm{d}\omega^{X}_t,
> $$
>
> where
>
> $$
> \omega^{X}_t = \int_0^t \sigma\bigl(\alpha \cdot X_s + \beta\bigr)\mathrm{d}s,
> $$
>
> and the trainable parameters are $\alpha \in \mathbb{R}^d$, $\beta \in \mathbb{R}$, and diagonal $D \in \mathbb{R}^{d_h \times d_h}$ [1]. It is possible to include the bias term in the Linear NCDE formulation, as shown in [1], but S6’s expressivity still remains limited. Furthermore, the block-diagonal, DPLR, Walsh–Hadamard, and Sparse SLiCEs are maximally expressive without a bias term, so it is not included in the SLiCE architecture. The other key difference with the S6 architecture is considering more general forms for the driving term $\omega^{X}_t$. In particular, all the experiments in this paper use
>
> $$
> \frac{\omega^X_{t_{j+1}}-\omega^X_{t_j}}{t_{j+1}-t_j} = (1, X_{t_{j}}).
> $$
>
> In contrast to models such as Mamba, which combines the S6 recurrence with a convolution and gated MLP, each SLiCE recurrence is combined with only:
>
> * a linear layer to mix the channels,
> * a tanh activation function,
> * layer normalisation, and
> * a skip connection.
>
> ---
>
> In addition to the above section, we have also included an algorithm outlining how to implement a structured linear CDE in practice.
>
> **Algorithm — Structured Linear CDE:**
> *The algorithm is presented for a dense state-transition matrix $A_{\theta}$. Comments indicate where the structure of $A_{\theta}$ can be used to reduce memory footprint and speed-up computation.*
>
> **Input:**  $\boldsymbol{\omega}$ : $(B, L, d_\omega)$
> **Output:** $\mathbf{h}$ : $(B, L, d_h)$
>
> 1. $\mathbf{h_0}$ : $(B, d_h)$ $\leftarrow$ $\xi_\phi(\boldsymbol{\omega_0})$
> 2. $\boldsymbol{\omega}^{\text{inc}}$ : $(B, L−1, d_\omega) \leftarrow \text{diff}(\boldsymbol{\omega})$
> 3. $A_{\theta}$ : $(d_\omega, d_h, d_h) \leftarrow$ *Parameter*  $\qquad\qquad\qquad\qquad\qquad\qquad\qquad$ # Exploit structure of $A_\theta$
> 4. $I$ : $(d_h, d_h) \leftarrow d_h\times d_h$ identity matrix
>
> 5. **if** *mode* = parallel **then**
> 6. $\quad F$ : $(B, L−1, d_h, d_h) \leftarrow I + \text{einsum}(bli, ijk \rightarrow bljk, \boldsymbol{\omega}^{\text{inc}}, A_\theta)$  # Broadcast $I$, exploit structure of $A_\theta$
> 7. $\quad F^{\text{comp}}$ : $(B, L−1, d_h, d_h) \leftarrow \text{pscan}(F)$ $\qquad\qquad\qquad\qquad\quad\ \ $ # Parallel associative scan, exploit structure of $F$
> 8. $\quad \mathbf{h_{1:L-1}}$ : $(B, L-1, d_h) \leftarrow \text{einsum}(bljk, bk → blj, F^{\text{comp}}, \mathbf{h_0})$
> 9. $\quad \mathbf{h}$ : $(B, L, d_h) \leftarrow [\mathbf{h_0}, \mathbf{h_{1:L-1}}]$
>
> 10. **else**  $\qquad\qquad\qquad\qquad\qquad\qquad\qquad\qquad\qquad\qquad\qquad\qquad\ \ \ $ # Recurrent pass
> 11. $\quad$**for** $t = 0$ **to** $L−2$:
> 12. $\quad \quad F_t$ : $(B, d_h, d_h) \leftarrow I + \text{einsum}(bi, ijk \rightarrow bjk,  \boldsymbol{\omega_t}^{\text{inc}}, A_\theta)$  $\qquad$ # Exploit structure of $A_\theta$
> 13. $ \quad \quad \mathbf{h_{t+1}}$ : $(B, d_h) \leftarrow \text{einsum}(bjk, bk \rightarrow bj, F_t, \mathbf{h_t})$ $\qquad\qquad\quad\ \ $ # Exploit structure of $F$
> 14. $\quad$**end for**
> 15. $\quad \mathbf{h}$ : $(B, L, d_h) \leftarrow [\mathbf{h_0}, …, \mathbf{h_{L−1}}]$ $\qquad\qquad\qquad\qquad\qquad\qquad\ \ \ $ # Stack along length axis
> 16. **end if**
> 17. **return** $\mathbf{h}$
>
> ---
>
> ## Weaknesses
>
> 1. We believe that the comment regarding missing implementation details stem from a misunderstanding of the simplicity of our proposed models. In particular, there are no multiplicative gates, head dimensions, or sharing structure to provide details on. We hope that our proposed additional section clarifies this, and that you now feel the implementation details for our models are sufficiently provided in Appendix D.
> 2. We hope that the proposed additional section has also clarified several of your questions here. The speed difference between the diagonal LNCDE and Mamba is partly due to the additional structure surrounding the S6 recurrence, but also due to different hyper-parameter choices, which were selected via a grid search. We have also expanded the results on the UEA dataset to include all of the SLiCE architectures, as detailed in our response to reviewer 7XXo under the heading "Questions 1, 3, and 4".
>
> Thank you for highlighting the distinction between S4 and S4D. We have updated our paper accordingly. Additionally, we have added a dedicated Limitations section to consolidate our discussion of constraints in one place.
>
> ---
>
> [1] Nicola Muca Cirone, Antonio Orvieto, Benjamin Walker, Cristopher Salvi, and Terry Lyons. *Theoretical foundations of deep selective state-space models.* Advances in Neural Information Processing Systems, 2024.

---

### Official Review · Reviewer_g3ZB · 2025-07-03

**Clarity:** 2
**Significance:** 3
**Originality:** 3
**Rating:** 6
**Confidence:** 3

**Summary:**

Technically dense paper. The denseness of the paper makes it slightly unapproachable for a non-expert. The paper introduces SLiCEs which are “Structured Linear Controlled Differential Equations”. The paper theoretically and experimentally shows the block-diagonal, sparse, and Walsh-hadamard SLiCEs are maximally probabilistically expressive. Empirically the paper validates many of its theoretical claims.

**Questions:**

1.How does performance degrade when mapping non-power-of-two dimensions before and after the transform? WHT requires a dimension which is a power-of-two – but curious to see performance outside of that. Please include results on one formal-language and one UEA dataset.
2. Could you evaluate BD-LNCDE and Log-NCDE on a synthetically irregularly subsampled version of one UEA dataset? Report accuracy and runtime (this would increase eval)
3. I fear many readers might find this paper technically dense, could you add a toy example to help readers get through the denser portions of the paper?

**Ethical Concerns:**

["NO or VERY MINOR ethics concerns only"]

**Final Justification:**

The authors did a great job assessing any concerns I had. I am still not extremely confident in the material - so I would be open to changing my score if a technical rebuttal was raised -- but for now I am happy with the paper.

**Limitations:**

Yes albeit scattered throughout the paper. Would always appreciate a dedicated section.

**Paper Formatting Concerns:**

Nothing Significant

**Quality:**

3

**Strengths And Weaknesses:**

Quality: The paper is theoretically rigorous and then provides decently robust empirical analysis.

Clarity: The paper did a good job of framing the role of SLiCEs as well as doing a good job with their figures. However the paper’s main weakness is the approachability. The paper could seriously benefit from some running toy examples to help cut through all the formalism.

Significance: The paper seems reasonably significant, both their theoretical results and practical results are good (20x speed up on per step time).

Originality: The sparse and Walsh SLiCEs and taxonomy seem quite novel.

---

> ### Author Rebuttal · Authors · 2025-07-31
>
> Thank you very much for your detailed review.
>
> ## Clarity
>
> We have redrafted the paper to improve the clarity. In particular, we have included the following two new lines in the introduction:
>
> 1. **After Line 19**: "Linear Neural Controlled Differential Equations (LNCDEs) are a continuous-in-time sequence model where the state-transition matrix, or vector field, depends linearly on the current input. This allows for the multiplicative interactions between the input and hidden state necessary for gating."
> 2. **After Line 28**: "SLiCEs replace the dense state-transition matrix of an LNCDE with an efficient structured variant, such as block-diagonal, sparse, Walsh–Hadamard, or DPLR, which maintain the maximal expressivity of dense matrices whilst significantly reducing both the parameter count and computational cost."
>
> Additionally, we have introduced a new chapter in the Appendix, which serves as a broad introduction to neural controlled differential equations. Topics covered include:
>
> * **Paths**, including bounded variation and discretisation/interpolation,
> * **Solving controlled differential equations**, including an expanded discussion on the discretised solve used in equation (2) (Lines 67-68),
> * **The signature and log-signature of a path**, which are crucial tools in the proofs of Theorems 4.1, 4.2, and 4.3 and the Log-ODE method.
>
> Furthermore, we have expanded our comparison to S6 and Mamba, by adding the following paragraph as Section 4.7 in the paper.
>
> ### Comparison to Existing Architectures
>
> SLiCEs are closely related to S6, the recurrent component of Mamba. In particular, if you remove the bias term (i.e. $B = 0$), then the continuous form of S6 is given by a LNCDE,
>
> $$
> dh_t = Dh_t \mathrm{d}\omega^{X}_t,
> $$
>
> where
>
> $$
> \omega^{X}_t = \int_0^t \sigma\bigl(\alpha \cdot X_s + \beta\bigr)\mathrm{d}s,
> $$
>
> and the trainable parameters are $\alpha \in \mathbb{R}^d$, $\beta \in \mathbb{R}$, and diagonal $D \in \mathbb{R}^{d_h \times d_h}$ [1]. It is possible to include the bias term in the Linear NCDE formulation, as shown in [1], but S6’s expressivity still remains limited. Furthermore, the block-diagonal, DPLR, Walsh–Hadamard, and Sparse SLiCEs are maximally expressive without a bias term, so it is not included in the SLiCE architecture. The other key difference with the S6 architecture is considering more general forms for the driving term $\omega^{X}_t$. In particular, all the experiments in this paper use
>
> $$
> \frac{\omega^X_{t_{j+1}}-\omega^X_{t_j}}{t_{j+1}-t_j} = (1, X_{t_{j}}).
> $$
>
> In contrast to models such as Mamba, which combines the S6 recurrence with a convolution and gated MLP, each SLiCE recurrence is combined with only:
>
> * a linear layer to mix the channels,
> * a tanh activation function,
> * layer normalisation, and
> * a skip connection.
>
> ## Toy Example
>
> We appreciate your recommendation for a toy example, and believe this will significantly benefit the approachability of the paper. Therefore, we have added the following motivating example as a demonstration of the difference in expressivity between models which do not mix their hidden-state channels (i.e. models with diagonal state-transition matrices) and models which do mix their hidden-state channels.
>
> Given a stream of bits
>
> $$
> x_1, x_2, \dots , x_n \in \\{0,1\\},
> $$
>
> we want to predict the parity label defined by
>
> $$
> p_n = S_n \bmod 2 \in \\{0,1\\},
> $$
>
> where
>
> $$
> S_n = \sum_{k=1}^n x_k .
> $$
>
> Whenever a new bit is \$1\$ the label flips; if the bit is \$0\$ the label stays the same.
> If we consider a diagonal LNCDE with a hidden dimension of \$2\$ and \$\omega^x\_{k+1}-\omega^x\_k = x\_{k+1}\$, then
>
> $$
> h_{n+1}
>   = \exp\left(
>       \begin{bmatrix}
>         a_1 & 0 \\\\
>         0   & a_2
>       \end{bmatrix}
>       x_{n+1}
>     \right)h_n,
> $$
>
>
> and
>
> $$
> h_n^{i} = h_0^{i}\exp\bigl(a_i S_n\bigr), \qquad i = 1,2 .
> $$
>
> With a linear read-out \$r=(r\_1,r\_2)^\top\$ followed by a monotone activation \$\phi\$ (such as \$\tanh\$, ReLU, sigmoid):
>
> $$
> \hat p_n
>   = \phi\Bigl(r^\top h_n\Bigr)
>   = \phi\Bigl(c_1 e^{a_1 S_n} + c_2 e^{a_2 S_n}\Bigr)
>   = \phi\Bigl(f(S_n)\Bigr),\qquad
>   c_i = r_i h_0^{\,i}.
> $$
>
> Since \$f(S)\$ has at most one turning point and \$\phi\$ is monotone, \$\hat{p}\_n\$ can cross any chosen threshold at most twice. However, the true label \$p\_n\$ flips every time \$S\_n \mapsto S\_n + 1\$. Hence, no diagonal \$2 \times 2\$ LNCDE can realise parity on arbitrarily long input.
>
> ---
>
> However, if you replace \$A\$ with
>
> $$
> A =
> \begin{pmatrix}
> 0 &  \pi \\\\
> -\pi & 0
> \end{pmatrix},
> $$
>
> then
>
> $$
> \exp(A x_{n+1})
>   = \begin{cases}
>       \begin{pmatrix}
>         1 & 0 \\\\
>         0 & 1
>       \end{pmatrix}, & \text{if } x_{n+1}=0, \\\\
>       \begin{pmatrix}
>        -1 & 0 \\\\
>         0 & -1
>       \end{pmatrix}, & \text{if } x_{n+1}=1.
>     \end{cases}
> $$
>
> Thus
>
> $$
> h_{n+1} = (-1)^{x_{n+1}} h_n
> \quad\Longrightarrow\quad
> h_n = (-1)^{S_n} h_0 .
> $$
>
> Taking \$r=(1,0)^\top\$, \$h\_0^{(1)}=1\$, and \$\phi(s)=\bigl(1-\operatorname{sign}(s)\bigr)/2\$,
>
> $$
> \hat p_n
>   = \frac{1 - \operatorname{sign}\bigl((-1)^{S_n}\bigr)}{2}
>   = S_n \bmod 2
>   = p_n.
> $$
>
> Hence, a dense \$2 \times 2\$ LNCDE can solve parity exactly with a hidden dimension of 2.
>
> ---
>
> ## Questions
>
> 1. Both the DPLR and diagonal-dense LNCDE use exclusively non-power-of-two dimensions for their hidden state on the regular-language tasks, and we do not see a major degradation in performance. In particular, the diagonal-dense LNCDE with a hidden dimension of \$272\$ achieves the highest average validation accuracy on the formal-language benchmark. For the UEA benchmarks, we specifically chose to match the hidden dimensions selected via the hyper-parameter grid search in \[1] in order to isolate the impact of the SLiCE structure from alternative hyper-parameter choices.
>
> 2. Results for S6, Log-NCDE, and BD-LNCDE on a synthetically irregularly subsampled version of EigenWorms (70 % of the samples dropped):
>
>    | Model    | EigenWorms (original) | EigenWorms (70 % dropped) |
>    | -------- | --------------------: | ------------------------: |
>    | S6       |     \$85.0 \pm 16.1\$ |         \$76.3 \pm 11.8\$ |
>    | Log-NCDE |      \$85.6 \pm 5.1\$ |          \$68.9 \pm 6.4\$ |
>    | BD-LNCDE |      \$86.7 \pm 4.1\$ |          \$85.0 \pm 2.8\$ |
>
>    *Table 1: Comparison of S6, Log-NCDE, and BD-LNCDE on irregularly subsampled EigenWorms.*
>
>    In addition to these results, we have significantly expanded our analysis of SLiCEs on the UEA datasets, including runtime comparisons. See our response to reviewer 7XXo for more details.
>
> 3. We have added the toy example outlined in the previous section.
>
> ---
>
> ## Limitations
>
> We have added a dedicated Limitations section to bring together our discussion of the constraints of this work in one place.
>
> ---
>
> [1] Benjamin Walker, Andrew D. McLeod, Tiexin Qin, Yichuan Cheng, Haoliang Li, and Terry Lyons. *Log Neural Controlled Differential Equations: The Lie Brackets Make a Difference.* International Conference on Machine Learning, 2024.

---

> > ### Comment · Reviewer_g3ZB · 2025-08-05
> > **Comment**
> >
> > Thanks so much for the comment. I appreciate you addressing all my concerns with clairty -- and adding the simple + effective toy example. I will be raising my score since there are no additional concerns from my end.

---

> > > ### Author Response · Authors · 2025-08-06
> > >
> > > Thank you for your thoughtful feedback and for taking the time to review our work. We’re glad to hear that our response and the toy example have addressed your concerns.

---

### Official Review · Reviewer_7XXo · 2025-07-08

**Clarity:** 2
**Significance:** 3
**Originality:** 3
**Rating:** 5
**Confidence:** 3

**Summary:**

This paper proposes SLiCEs, a framework that unifies sequence-to-sequence layers that are maximally expressive, computationally efficient, and allow for parallel-in-time computation. This framework ties together existing architectures, such as input-dependent block-diagonal linear RNNs and diagonal-plus-low-rank structure, and new variants based on sparsity and the Walsh-Hadamard transform. The authors show that these structured sequence models have maximal probabilistic expressivity and test them on various tasks, including the A5 and regular language tasks along with datasets from the UEA multivariate time-series classification archive.

**Questions:**

The paper notes that DPLR, sparse, and WH-LNCDEs are not closed under multiplication, which means that the limiting computational cost per composition is the same as a dense LNCDE. If I understand correctly, this is one of the reasons why diagonal SSMs have been used. In practice, how do the runtimes of diagonal, DPLR, sparse, BD, dense, WH-LNCDEs compare? (e.g., the average time / 1k steps (s) as reported in Table 3).

Related to the question above, why does log-NCDE take orders of magnitude longer in terms of average time / 1k steps (s) in Table 3?

In Table 3, I suppose that it was infeasible to train the sparse model due to the lack of an efficient implementation. How does the WH-LNCDE model perform on this set of datasets?

On lines 219-220, the paper introduces a hybrid strategy, where the Log-ODE method is applied to small intervals, and the resulting outputs are then processed using a parallel associative scan. In practice, how does the choice of interval affect the runtime and memory cost?

Related to line 298, have you tried applying various regularization techniques to DE-LNCDE? It does seem odd that DE-LNCDE performs worse than BD-LNCDE, almost being on par with D-LNCDE.

For the formal language tasks, why does WH-LNCDE underperform other models that equally have maximal probabilistic expressivity? Do all models have the same number of parameters? Would increasing the state size of WH-LNCDE help its performance match other models? If so, by how much?

On lines 274-275, the paper notes that diagonal LNCDE outperforms Mamba since it includes a wider range of eigenvalues. However, the result on Mod Arith. No Brack shows the opposite. Do you have an intuition on why that’s the case?

I wonder how different LNCDE models do on generalizing to longer sequences for the A5 task, as in the regular language task. This experiment is of interest in the community, as shown in [1].

[1] Grazzi, Riccardo, et al. "Unlocking state-tracking in linear rnns through negative eigenvalues." arXiv preprint arXiv:2411.12537 (2024).

**Ethical Concerns:**

["NO or VERY MINOR ethics concerns only"]

**Final Justification:**

Through the rebuttal process, the authors addressed my main concerns. I thus decided to increase the score accordingly.

**Limitations:**

Yes.

**Paper Formatting Concerns:**

I did not notice any major formatting issues in this paper.

**Quality:**

3

**Strengths And Weaknesses:**

Quality: The submission is technically sound. Under the SLiCEs framework, the authors rigorously show that various existing and novel architectures have maximal probabilistic expressivity. The methods are tested on various state tracking tasks, which are great benchmark datasets to test the expressivity of stateful sequence models. However, there are several points that I think would be helpful if the authors could address (in the Questions section)

Clarity: Overall, the paper does a great job introducing relevant concepts and technical details to the readers. The experimental details are clear, with additional details written in the appendix. However, I believe that the abstract and the introduction could be improved to better convey the overall message of the paper. For example, just by looking at the abstract, it is unclear whether SLiCEs is an existing or novel framework. In addition, I think adding a couple of more sentences in the introduction to describe LNCDEs and how exactly SLiCEs extend LNCDEs would make it flow better.

Significance: Designing expressive stateful recurrent models is of great interest to the ML community. Thus, the paper’s framework that ties together various architectures, along with a couple of novel structures, will greatly help the community. The paper’s empirical results on the superior performance of block-diagonal models over others are also interesting.

Originality: The framework proposed by the paper deepens our understanding of the existing structured recurrent models. In addition, to the best of my knowledge, utilizing the Walsh-Hadamard structure on state transitions is novel and worth further investigation.

---

> ### Author Rebuttal · Authors · 2025-07-31
>
> Thank you very much for your thoughtful review.
>
> ## Clarity
>
> We have redrafted the abstract and introduction to improve the clarity. Key edits include:
>
> 1. **Line 1**: "This work introduces Structured Linear Controlled Differential Equations (SLiCEs), a unifying framework"
> 2. **After Line 19**: "Linear Neural Controlled Differential Equations (LNCDEs) are a continuous-in-time sequence model where the state-transition matrix, or vector field, depends linearly on the current input. This allows for the multiplicative interactions between the input and hidden state necessary for gating."
> 3. **After Line 28**: "SLiCEs replace the dense state-transition matrix of an LNCDE with an efficient structured variant, such as block-diagonal, sparse, Walsh–Hadamard, or DPLR, which maintain the maximal expressivity of dense matrices whilst significantly reducing both the parameter count and computational cost."
> 4. **Line 30**: "We introduce Structured Linear Controlled Differential Equations (SLiCEs)"
>
> ## Questions
>
> ### Questions 1, 3, and 4
>
> We now evaluate all SLiCEs on the UEA benchmark. The D-DE and DPLR structure use a dense block of dimension $16$ and a rank of $4$, respectively, as these were the best-performing choices on the formal-language benchmarks. BD-LNCDE is the best-performing, followed by Log-NCDE, DE-LNCDE, and WH-LNCDE. We find that D-DE, S, and DPLR LNCDEs underperform S5, S6, LRU, and a D-LNCDE. Together with the formal-language tasks, these findings indicate that BD provides practical benefits relative to the other SLiCE structures considered. In particular, we believe the connection with multi-head attention (Lines 169-170) is important, and have redrafted our paper to expand this discussion.
>
> | Model  | EW   | EC | HB  | MI  | SCP1  | SCP2  | Av. Acc  | Av. Rank |
> | ---------- | -------------- | -------------- | -------------- | -------------- | -------------- | -------------- | -------- | -------- |
> | BD-LNCDE | 86.7 ± 4.1     | 28.6 ± 6.4     | 75.2 ± 6.0     | **58.3 ± 4.1** | 84.9 ± 1.9   | 53.3 ± 7.5   | **64.5** | **3.5**  |
> | Log-NCDE | 85.6 ± 5.1     | **34.4 ± 6.4** | 75.2 ± 4.6     | 53.7 ± 5.3     | 83.1 ± 2.8     | **53.7 ± 4.1** | 64.3   | 3.8    |
> | DE-LNCDE | 87.8 ± 5.8   | 25.6 ± 3.7     | 76.1 ± 3.1     | 54.0 ± 5.3   | 83.5 ± 6.6     | 46.3 ± 3.6     | 62.2     | 6.0      |
> | WH-LNCDE | 83.9 ± 7.3     | 29.4 ± 4.0     | 75.2 ± 5.7     | 50.5 ± 4.2     | 82.4 ± 2.1     | 51.2 ± 4.7     | 62.1     | 6.5      |
> | S6  | 85.0 ± 16.1    | 26.4 ± 6.4     | 76.5 ± 8.3     | 51.3 ± 4.7     | 82.8 ± 2.7     | 49.9 ± 9.5     | 62.0     | 7.0      |
> | LRU | 87.8 ± 2.8   | 21.5 ± 2.1     | **78.4 ± 6.7** | 48.4 ± 5.0     | 82.6 ± 3.4     | 51.2 ± 3.6     | 61.7     | 7.0      |
> | D-LNCDE | 80.0 ± 5.4     | 25.8 ± 4.0     | 72.9 ± 5.0     | 54.0 ± 7.3   | 83.5 ± 2.1     | 53.0 ± 5.8     | 61.6     | 7.2      |
> | S5  | 81.1 ± 3.7     | 24.1 ± 4.3     | 77.7 ± 5.5   | 47.7 ± 5.5     | **89.9 ± 4.6** | 50.5 ± 2.6     | 61.8     | 7.8      |
> | D-DE-LNCDE | **88.9 ± 6.1** | 27.6 ± 2.9     | 74.8 ± 3.3     | 47.7 ± 2.8     | 81.6 ± 4.9     | 50.9 ± 4.6     | 61.9     | 8.0      |
> | S-LNCDE | 82.2 ± 2.8     | 29.1 ± 6.6     | 73.2 ± 4.7     | 50.5 ± 3.5     | 81.4 ± 0.9     | 52.3 ± 4.1     | 61.5     | 8.3      |
> | DPLR-LNCDE | 77.8 ± 6.8     | 28.1 ± 3.7     | 73.9 ± 5.1     | 54.0 ± 6.4   | 82.8 ± 4.6     | 45.6 ± 5.0     | 60.4     | 8.3      |
> | NCDE | 75.0 ± 3.9     | 29.9 ± 6.5   | 73.9 ± 2.6     | 49.5 ± 2.8     | 79.8 ± 5.6     | 53.0 ± 2.8     | 60.2     | 8.7      |
> | NRDE | 83.9 ± 7.3     | 25.3 ± 1.8     | 72.9 ± 4.8     | 47.0 ± 5.7     | 80.9 ± 2.5     | **53.7 ± 6.9** | 60.6     | 9.8      |
> | MAMBA | 70.9 ± 15.8    | 27.9 ± 4.5     | 76.2 ± 3.8     | 47.7 ± 4.5     | 80.7 ± 1.4     | 48.2 ± 3.9     | 58.6     | 10.5     |
> *UEA test accuracy*
>
> Additionally, we have run a number of experiments comparing the time per $1000$ training steps for the different SLiCE structures on EigenWorms ($\approx 18{,}000$ observations). When computing recurrently, going from DE to D, BD, or WH reduces the run-time. Note DPLR actually has a longer runtime than DE. This is likely due to our implementation incurring the computational overhead of three separate operations, which outweighs the benefit of the reduced number of FLOPS. For the D and BD LNCDE, applying the parallel associative scan strictly decreases run time, as expected. Although the same is theoretically true for DE-LNCDE, the I/O costs are magnified for DE matrices, making the associative scan slower than recurrent computation when using a small number of parallel steps. Since WH and DPLR are treated as DE when applying a parallel associative scan, they follow the same behaviour. We did not include S in the comparison as it is always treated as DE.
>
> | Parallel Steps | D-LNCDE | BD-LNCDE | WH-LNCDE | DPLR-LNCDE | DE-LNCDE |
> | -------------: | ------: | -------: | -------: | ---------: | -------: |
> |   None |  213.45 |   226.59 |   242.49 |     298.67 |   279.31 |
> |      4 |   92.87 |   179.94 |   396.25 |     399.98 |   406.56 |
> |     16 |   38.56 |    91.15 |   245.04 |     245.55 |   244.60 |
> |     64 |   19.48 |    39.53 |   124.75 |     123.84 |   123.73 |
> |    256 |   11.15 |    19.98 |    95.86 |      94.66 |    90.46 |
>
> *Training time per 1 k steps (s)*
>
> We also compare runtime and GPU memory on EigenWorms when using the Log-ODE method. See our response to reviewer EATe for a theoretical computational cost analysis. The additional memory for BD-LNCDE is negligible. However, for WH-LNCDE, where the matrices are treated as dense, the Log-ODE method incurs significant additional memory. For both models, the Log-ODE method leads to a significant decrease in runtime. A parallel associative scan leads to further runtime reductions for the BD-LNCDE. However, for the WH-LNCDE, the parallel associative scan incurs too high a GPU memory cost when not using the Log-ODE method, and is slower than recurrent calculation when using the Log-ODE method. This is due to the high I/O costs when using dense matrices.
>
> | Metric                    | Log-ODE Interval | Recurrent | Parallel steps 128 |
> | ------------------------- | ---------------- | ---------: | -----------: |
> | **Time / 1 k steps [s]** | 1                |     227.26 |        52.58 |
> |                           | 12               |      24.66 |        13.40 |
> | **GPU Memory [GB]**      | 1                |       3.48 |         7.59 |
> |                           | 12               |       3.48 |         3.49 |
>
> *BD-LNCDE*
>
> | Metric                    | Log-ODE Interval | Recurrent | Parallel steps 128 |
> | ------------------------- | ---------------- | ---------: | -----------: |
> | **Time / 1 k steps [s]** | 1                |     450.41 |          OOM |
> |                           | 12               |      55.46 |       135.52 |
> | **GPU Memory [GB]**      | 1                |       3.48 |          OOM |
> |                           | 12               |      13.72 |        18.58 |
>
> *WH-LNCDE*
>
> ### Question 2
>
> Non-linear NCDEs numerically solve a non-linear differential equation, which is costly and recurrent.
>
> ### Question 5
>
> All SLiCEs use weight regularisation on their state-transition matrices. The paper did not make this explicit and we apologise for this oversight; the main body and Appendix now discuss it. In addition, due to training-stability issues, we constrained the entries of the Walsh–Hadamard LNCDE’s diagonal matrix to be in the range $[-1,1]$ (Lines 808-809, 856-857), restricting the eigenvalues of the state-transition matrix to modulus $\leq 1$. A broader investigation of regularisation techniques is important future work but sits outside the scope of this paper.
>
> ### Question 6
>
> All SLiCEs were initially evaluated on the formal-language tasks using the same number of non-zero parameters ($512$) in their state-transition matrices (Line 264). This yields a wide range of total parameter counts, owing to linear layers between SLiCE layers and varying hidden dimension with varying block sizes/ranks. To mitigate hidden-dimension effects, the WH, D-LNCDE, and all baselines were also run with hidden dimension $128$, with the best choice selected for each dataset (Lines 259, 265, 835, 851). We have clarified this motivation. In practice, WH-LNCDE performed better on some tasks with hidden dimension $128$ (cycle navigation, even pairs) and others with $512$ (modular arithmetic without brackets, parity). Overall average validation accuracies were $44.9$ (dim 128) and $45.2$ (dim 512).
>
> ### Question 7
>
> We have added a detailed comparison between D-LNCDEs and Mamba. See our response to reviewer oajU under the heading "Clarity" for full details. In brief, Mamba combines the S6 recurrence, a convolution, and a gated MLP, whereas LNCDEs are a recurrence followed by a non-linear channel mixing. We added S6 as a baseline on the formal-language tasks; it fails to solve modular arithmetic without brackets, suggesting that Mamba’s additional components are what enable performance above random guessing.
>
> | Model | Cycle Nav. |  Even Pairs | Mod Arith.  |     Parity | Average |
> | ----- | ---------: | ----------: | ---------------------: | ---------: | ------: |
> | S6    | 33.9 ± 1.0 | 100.0 ± 0.0 |  21.4 ± 1.2 | 50.8 ± 0.4 |    51.5 |
>
> ### Question 8
>
> We are adding a length-generalisation experiment on the A5 benchmark. Models are trained on sequences of lengths 3–40, with early stopping on a validation set of sequences of lengths 40–128. Preliminary results show nearly 90 % token accuracy for sequences up to 16 × longer than those seen in training.
>
> | Model    |     40 |     80 |   160 |   320 |   640 |  1280 |  2560 |
> | -------- | -----: | -----: | ----: | ----: | ----: | ----: | ----: |
> | S-LNCDE  | 100.00 | 100.00 | 99.95 | 99.06 | 94.33 | 78.32 | 47.60 |
> | DE-LNCDE |  99.99 |  99.98 | 99.99 | 99.99 | 91.86 | 53.52 | 29.27 |
> | BD-LNCDE | 100.00 | 100.00 | 99.49 | 95.95 | 89.41 | 70.07 | 39.06 |

---

> > ### Comment · Reviewer_7XXo · 2025-08-04
> >
> > I thank the authors for their detailed response. I will adjust the score accordingly.

---

> > > ### Author Response · Authors · 2025-08-05
> > >
> > > Dear Reviewer 7XXo,
> > >
> > > Thank you for your thoughtful consideration of our rebuttal and for adjusting your score. We are pleased that our response has addressed the points you raised. Should any concerns remain, we would be glad to provide further clarification.

---

### Note · Authors · 2025-08-14

We thank the AC and all reviewers for their careful reading, constructive feedback, and insightful suggestions. The guidance we received has led to notable improvements in both the clarity and scope of the paper. Below we summarise the key changes.

### Clarity and positioning
- We rewrote the abstract and introduction to improve clarity around LNCDEs and to clearly state that SLiCEs are a novel extension that unifies multiple structured, input-dependent, state-transition matrices.
- We added an accessible toy example on parity which illustrates why mixing hidden-state channels matters for expressivity.
- We inserted a new section comparing SLiCEs to existing architectures (S6/Mamba, DeltaNet, etc.), highlighting similarities and differences in both mathematics and the surrounding block structure.
- We included a step-by-step algorithm for using structured linear CDEs in practice.

### Expanded empirical evaluation
- We extended the UEA-MTSCA evaluation to include all SLiCE structures, and we report the effect of using a parallel associative scan and the Log-ODE method on both runtime and GPU memory. In particular, BD-LNCDE’s state-transition matrix maintains its structure under composition, allowing it to benefit strongly from both the Log-ODE method and the parallel associative scan.
- We added a length-generalisation study on $A_5$, with training on lengths $3$–$40$, early stopping on $40$–$128$, and testing on lengths up to $2{,}560$. SLiCE variants maintain almost $90\\%$ token accuracy on sequences up to $16\times$ longer than those seen during training.

We believe these revisions, alongside the targeted changes made in response to individual reviewer comments, have addressed the reviewers' concerns and strengthened the paper. We again thank the AC and all reviewers for their valuable feedback.

---

### Decision · Program_Chairs · 2025-09-17

**Decision:**

Accept (spotlight)

**Comment:**

All reviewers valued the technical contributions of the SLiCEs framework, how it unifies several existing deep SSM architectures, and the avenues it opens for new designs. While the manuscript is technically challenging for non-experts, the authors have taken steps to make it more approachable. I congratulate the authors on their nice work — it is a valuable and timely contribution to the field.